# Scalable First-order Method for Certifying Optimal k-Sparse GLMs

Jiachang Liu [1]  Soroosh Shafiee [1]  Andrea Lodi [2]

## Abstract

This paper investigates the problem of certifying optimality for sparse generalized linear models (GLMs), where sparsity is enforced through an $\ell_0$ cardinality constraint. While branch-and-bound (BnB) frameworks can certify optimality by pruning nodes using dual bounds, existing methods for computing these bounds are either computationally intensive or exhibit slow convergence, limiting their scalability to large-scale problems. To address this challenge, we propose a first-order proximal gradient algorithm designed to solve the perspective relaxation of the problem within a BnB framework. Specifically, we formulate the relaxed problem as a composite optimization problem and demonstrate that the proximal operator of the non-smooth component can be computed exactly in log-linear time complexity, eliminating the need to solve a computationally expensive second-order cone program. Furthermore, we introduce a simple restart strategy that enhances convergence speed while maintaining low per-iteration complexity. Extensive experiments on synthetic and real-world datasets show that our approach significantly accelerates dual bound computations and is highly effective in providing optimality certificates for large-scale problems.

## 1. Introduction

Sparse generalized linear models (GLMs) are essential tools in machine learning (ML), widely applied in fields like healthcare, finance, engineering, and science. These models provide a flexible framework for capturing relationships between variables while ensuring interpretability, which

[1]School of Operations Research and Information Engineering, Cornell University, Ithaca, NY, USA [2]Jacobs Technion-Cornell Institute, Cornell Tech and Technion–IIT, New York, NY, USA. Correspondence to: Jiachang Liu <jiachang.liu@cornell.edu>, Soroosh Shafiee <shafiee@cornell.edu>, Andrea Lodi <al748@cornell.edu>.

*Proceedings of the 42nd International Conference on Machine Learning*, Vancouver, Canada. PMLR 267, 2025. Copyright 2025 by the author(s).

is critical in high-stakes applications. Recently, using the $\ell_0$ norm to induce sparsity has gained significant attention. This approach provides distinct advantages over traditional convex relaxation methods, such as replacing $\ell_0$ with $\ell_1$, particularly in cases involving highly correlated features.

In this paper, we aim to solve

$$
\begin{aligned}
\min_{\boldsymbol{\beta} \in \mathbb{R}^p} \quad & f(\boldsymbol{X}\boldsymbol{\beta}, \boldsymbol{y}) + \lambda_2 \|\boldsymbol{\beta}\|_2^2 \\
\text{s.t.} \quad & \|\boldsymbol{\beta}\|_\infty \leq M, \ \|\boldsymbol{\beta}\|_0 \leq k,
\end{aligned}
\tag{1}
$$

where $\boldsymbol{X} \in \mathbb{R}^{n \times p}$ and $\boldsymbol{y} \in \mathbb{R}^n$ denote the matrix of features and the vector of labels, respectively, while the parameter $M > 0$ can be either user-defined based on prior knowledge or estimated from the data (Park & Klabjan, 2020). The GLM loss function, denoted by $f : \mathbb{R}^n \times \mathbb{R}^n \to \mathbb{R}$, is assumed to be Lipschitz smooth, the parameter $k \in \mathbb{N}$ controls the number of nonzero coefficients, and $\lambda_2 > 0$ is a small Tikhonov regularization coefficient to address collinearity. However, problem (1) is NP-hard (Natarajan, 1995). As a result, most existing methods rely on heuristics that deliver high-quality approximations but lack guarantees of optimality. This limitation is particularly problematic in high-stakes applications like healthcare, where ensuring accuracy, reliability, and safety is essential. Therefore, we emphasize the pursuit of certifiably optimal solutions.

A naive approach to solve (1) to optimality is to reformulate it as a mixed-integer programming (MIP) problem and use commercial MIP solvers. However, these solvers struggle with scalability, especially for large datasets or nonlinear objectives. A key bottleneck is the computation of tight lower bounds at each branch-and-bound (BnB) node, which is essential for effective pruning and solver performance. Existing methods for computing lower bounds typically rely on linear or conic optimization. However, these approaches either generate loose bounds that reduce pruning efficiency or result in high computational costs per iteration. Additionally, they are difficult to parallelize, limiting their ability to leverage modern hardware accelerators like GPUs.

To address these challenges, we propose a scalable first-order method for efficiently calculating lower bounds within the BnB framework. We begin with a perspective reformulation of (1) and derive its continuous relaxation. The resulting formulation is then expressed as an unconstrained optimization problem, characterized by a convex compos-

ite objective function, which enables the application of the Fast Iterative Shrinkage-Thresholding Algorithm (FISTA), a well-known first-order method (Beck & Teboulle, 2009), to compute lower bounds. The successful implementation of FISTA, however, relies on efficient computation of the proximal operator, which requires solving a second order cone program (SOCP) problem. To the best of our knowledge, the efficient computation of this proximal operator has not been previously addressed in the literature. Therefore, we propose a customized pooled-adjacent-violation algorithm (PAVA) that evaluates the proximal operator exactly with log-linear time complexity, ensuring the scalability of our FISTA approach for large problem instances. A major advantage of our approach is its computational efficiency, in which instead of solving costly linear systems, it only relies on matrix-vector multiplication, which is highly amenable to GPU acceleration. This capability addresses a key limitation of existing approaches that struggle to parallelize their computations on modern hardware.

To accelerate the performance of the FISTA algorithm, we introduce a restart heuristic. This leads to an empirical linear convergence rate, a result not previously achieved by other first-order methods for this type of problem. Empirically, our method demonstrates substantial speedups in computing dual bounds – often by 1-2 orders of magnitude – compared to existing techniques. These improvements significantly enhance the overall efficiency of the BnB process, enabling the certification of large-scale instances of (1) that were previously intractable using commercial MIP solvers. All omitted proofs are provided in Appendix A. The experimental setup is detailed in Appendix B. Additional numerical results are reported in Appendix C.

### 1.1. Contributions

The key contributions of this paper are summarized below.

◇ We propose a FISTA-based first-order method to enhance the scalability of solving (1), with a focus on efficient lower-bound computation within the BnB framework.

◇ The proximal operator in the FISTA method is computed using a customized PAVA that leverages hidden mathematical structures and enjoys log-linear time complexity, ensuring scalability for large-scale problems.

◇ Besides achieving fast convergence rates (via a restart strategy) and low per-iteration computational complexity, our method can be easily parallelized on GPUs, something not currently achievable by MIP methods.

◇ We validate the practical efficiency of our approach on both synthetic and real-world datasets, demonstrating substantial speedups in computing dual bounds and certifying optimal solutions for large-scale sparse GLMs.

### 1.2. Related Works

**MIP for ML.** MIP has been successfully applied in medical scoring systems (Ustun & Rudin, 2016; 2019; Liu et al., 2022), portfolio optimization (Bienstock, 1996; Wei et al., 2024), nonlinear identification systems (Bertsimas & Gurnee, 2023; Liu et al., 2024), decision trees (Bertsimas & Dunn, 2017; Hu et al., 2019), survival analysis (Zhang et al., 2023; Liu et al., 2025), hierarchical models (Bertsimas & Van Parys, 2020a), regression and classification models (Atamturk & Gómez, 2020; Bertsimas & Van Parys, 2020a; Bertsimas et al., 2020; Bertsimas & Van Parys, 2020b; Hazimeh & Mazumder, 2020; Xie & Deng, 2020; Atamtürk et al., 2021; Dedieu et al., 2021; Hazimeh et al., 2022; Liu et al., 2024; Guyard et al., 2024), graphical models (Manzour et al., 2021; Kucukyavuz et al., 2023), and outlier detection (Gómez, 2021; Gómez & Neto, 2023). The primary focus of these works is on obtaining high-quality feasible solutions, with only a small subset addressing the certification of optimality. Our work aims to contribute to this literature, with a strong focus on enhancing the computational scalability of certifying optimality for solving sparse GLM problems.

**Perspective Formulations.** The application of perspective functions to derive convex relaxations for (1) dates back to the seminal work of Ceria & Soares (1999). Perspective formulations have been developed for separable functions in (Günlük & Linderoth, 2010; Xie & Deng, 2020; Wei et al., 2022; Cacciola et al., 2023; Bacci et al., 2024; Shafiee & Kılınç-Karzan, 2024) and for rank-one functions in (Frangioni et al., 2020; Wei et al., 2020; 2022; Atamtürk & Gómez, 2023; Han & Gómez, 2024; Shafiee & Kılınç-Karzan, 2024) under various conditions. Our work uses perspective formulation of separable functions that appear in (1) as the Tikhonov regularization function. This formulation provides tighter relaxations compared to the $\ell_1$-based methods proposed in (Mhenni et al., 2020).

**Lower Bound Calculation.** A key aspect of certifying optimality in MIP problems is the efficient computation of tight lower bounds. Commercial MIP solvers typically iteratively linearize the objective function using the celebrated outer approximation method (Kelley, 1960) (via cutting planes) and solve the resulting linear program (Schrijver, 1998; Wolsey, 2020). However, this approach often produce loose lower bounds, especially when high-quality linear cuts are not generated. Alternatively, solvers may use conic convex relaxations and solve them with the interior-point method (IPM) (Dikin, 1967; Renegar, 2001; Nesterov & Nemirovskii, 1994). While this approach often yields tighter lower bounds, IPM does not scale well due to its reliance on second-order information and because – differently from the linear case – effectively warm-starting IPMs is not possible.

Recent attempts are based on first-order methods, including subgradient descent (Bertsimas et al., 2020), ADMM (Liu et al., 2024), and coordinate descent (Hazimeh et al., 2022). Our work builds on this, offering faster convergence, low computational complexity, and significant GPU acceleration. We also observe that our proposed FISTA method achieves linear convergence rates empirically, a result not previously achieved by other first-order methods for this problem.

**GPU Acceleration.** Recently, there have been some promising works on using GPUs to accelerate continuous optimization problems, including linear programming (Applegate et al., 2021; Lu et al., 2023), quadratic programming (Lu & Yang, 2023), and semidefinite programming (Han et al., 2024). A natural way to leverage GPUs for discrete problems is to implement greedy heuristics on GPUs (Blanchard & Tanner, 2013). Alternatively, one can use GPU-based LPs within MIP solvers, as demonstrated by De Rosa et al. (2024) for solving clustering problems. However, in (De Rosa et al., 2024), the challenge is to approximate the original objective function with a potentially exponential number of cutting planes. In contrast, we develop a customized FISTA method that directly handles the nonlinear objective function, while the computation can be easily parallelized since it only involves matrix-vector multiplication. Other first-order methods, such as ADMM (Liu et al., 2024) and coordinate descent (Hazimeh et al., 2022), are unsuitable for GPUs: ADMM requires solving linear systems, while coordinate descent is inherently sequential.

## 2. Problem Formulation

In this preliminary section, we introduce some backgrounds on how to obtain a lower bound (which will be used for the branch-and-bound process to prune nodes) for the optimal value of Problem (1) by solving an associated convex relaxation problem. First, note that we can cast problem (1) as

$$\min \left\{ \tau \,:\, (\tau, \boldsymbol{\beta}, \boldsymbol{z}) \in \mathcal{S} \right\}, \qquad (2)$$

where the extended feasible set is defined as

$$\mathcal{S} = \left\{ (\tau, \boldsymbol{\beta}, \boldsymbol{z}) \,\middle|\, \begin{array}{l} \|\boldsymbol{\beta}\|_\infty \leq M, \\ \boldsymbol{z} \in \{0,1\}^p,\, \mathbf{1}^T \boldsymbol{z} \leq k, \\ \beta_j(1-z_j) = 0 \;\forall j \in [p], \\ f(\boldsymbol{X}\boldsymbol{\beta}, \boldsymbol{y}) + \lambda_2 \|\boldsymbol{\beta}\|_2^2 \leq \tau \end{array} \right\}, \qquad (3)$$

and $[p] = \{1, \dots, p\}$ stands for the set of all integers up to $p \in \mathbb{N}$. Put it differently, each binary variable $z_j$ indicates whether a continuous variable $\beta_j$ is zero or not by requiring $\beta_j = 0$ when $z_j = 0$ and allowing $\beta_j$ to take any value when $z_j = 1$. Meanwhile, the objective function is linearized using the epigraph reformulation technique, which allows us to interpret the optimal value of (2) as the evaluation of the support function of $\mathcal{S}$ at $(\mathbf{0}, \mathbf{0}, 1)$. By virtue of (Rockafellar,

1970, §13), the optimal value of (2) remains unchanged if we replace $\mathcal{S}$ with $\mathrm{cl\,conv}(\mathcal{S})$, where $\mathrm{cl\,conv}(\mathcal{S})$ denotes the closed convex hull of $\mathcal{S}$. However, the exact description of $\mathrm{cl\,conv}(\mathcal{S})$ requires exponentially many (nonlinear) constraints, which leads to the NP-hardness of (1).

We thus explore other options for a convex relaxation of (1). It turns out that a tractable convex hull can be obtained if the objective function only includes the Tikhonov regularization term $\|\boldsymbol{\beta}\|_2^2$, using the perspective function. The perspective function of the quadratic function $h(\beta) = \beta^2$ is $h^\pi(\beta, z) = \beta^2/z$ if $z > 0$, $= 0$ if $\beta = z = 0$, and $= \infty$ otherwise. For simplicity, we write $\beta^2/z$ instead of $h^\pi(\beta, z)$ even if $z = 0$. The following lemma provides an exact perspective formulation of the convex hull when $\mathcal{S}$ does not include $f(\boldsymbol{X}\boldsymbol{\beta}, \boldsymbol{y})$. This result extends (Günlük & Linderoth, 2010, Lemma 6) by incorporating sparsity constraints, while also extending (Shafiee & Kılınç-Karzan, 2024, Theorem 2) to account for $\ell_\infty$ box constraint on $\boldsymbol{\beta}$.

**Lemma 2.1.** *The closed convex hull of the set*

$$\left\{ (\tau, \boldsymbol{\beta}, \boldsymbol{z}) \,\middle|\, \begin{array}{l} \|\boldsymbol{\beta}\|_\infty \leq M, \\ \boldsymbol{z} \in \{0,1\}^p,\, \mathbf{1}^T \boldsymbol{z} \leq k, \\ \beta_j(1-z_j) = 0 \;\forall j \in [p], \\ \sum_{j \in [p]} \beta_j^2 \leq \tau \end{array} \right\}$$

*is given by the set*

$$\left\{ (\tau, \boldsymbol{\beta}, \boldsymbol{z}) \,\middle|\, \begin{array}{l} -Mz_j \leq \beta_j \leq Mz_j \;\forall j \in [p], \\ \boldsymbol{z} \in [0,1]^p,\, \mathbf{1}^T \boldsymbol{z} \leq k, \\ \sum_{j \in [p]} \beta_j^2/z_j \leq \tau \end{array} \right\}.$$

The convex hull formulation presented in Lemma 2.1 is a second-order conic set. Specifically, the epigraph of the sum of perspective functions in the last line satisfies

$$\sum_{j \in [p]} \beta_j^2/z_j \leq \tau \iff \exists \boldsymbol{t} \in \mathbb{R}_+^p \;\text{s.t.}\; \begin{cases} \mathbf{1}^\top \boldsymbol{t} = \tau, \\ \beta_j^2 \leq z_j t_j \;\forall j \in [p], \end{cases}$$

which is second order cone representable. Motivated by Lemma 2.1, we immediately see that the extended feasible set $\mathcal{S}$ defined in (3) admits the following perspective representation

$$\mathcal{S} = \left\{ (\tau, \boldsymbol{\beta}, \boldsymbol{z}) \,\middle|\, \begin{array}{l} -Mz_j \leq \beta_j \leq Mz_j \; j \in [p], \\ \boldsymbol{z} \in \{0,1\}^p,\, \mathbf{1}^T \boldsymbol{z} \leq k, \\ f(\boldsymbol{X}\boldsymbol{\beta}, \boldsymbol{y}) + \lambda_2 \sum_{j \in [p]} \beta_j^2/z_j \leq \tau \end{array} \right\}.$$

Plutting in this new perspective representation into Problem (2), we can reformulate (1) as follows

$$P_{\mathrm{MIP}}^\star = \begin{cases} \displaystyle\min_{\boldsymbol{\beta}, \boldsymbol{z} \in \mathbb{R}^p} & f(\boldsymbol{X}\boldsymbol{\beta}, \boldsymbol{y}) + \lambda_2 \sum_{j \in [p]} \beta_j^2/z_j \\ \text{s.t.} & \boldsymbol{z} \in \{0,1\}^p,\, \mathbf{1}^T \boldsymbol{z} \leq k, \\ & -Mz_j \leq \beta_j \leq Mz_j \;\forall j \in [p]. \end{cases} \qquad (4)$$

By relaxing the binary variables $z_j$ to the interval $[0, 1]$, we obtain the following strong convex relaxation of (4)

$$P_{\text{conv}}^\star = \begin{cases} \min_{\boldsymbol{\beta}, \boldsymbol{z} \in \mathbb{R}^p} & f(\boldsymbol{X}\boldsymbol{\beta}, \boldsymbol{y}) + \lambda_2 \sum_{j \in [p]} \beta_j^2 / z_j \\ \text{s.t.} & \boldsymbol{z} \in [0, 1]^p, \, \mathbf{1}^T \boldsymbol{z} \le k, \\ & -M z_j \le \beta_j \le M z_j \, \forall j \in [p]. \end{cases} \quad (5)$$

Although this is not the convex hull formulation due to the term $f(\boldsymbol{X}\boldsymbol{\beta}, \boldsymbol{y})$, unlike in Lemma 2.1, $P_{\text{conv}}^\star$ still provides a lower bound for Problem (1).

We can solve (5) using standard conic optimization solvers like Mosek and Gurobi, which rely on IPMs for solving such subproblems in the BnB framework. However, IPMs are computationally expensive and do not scale well for large datasets. Alternatively, first-order conic solvers such as SCS (O'donoghue et al., 2016), based on ADMM, can be used. While these methods are more scalable, they suffer from slow convergence rates and require solving linear systems at each iteration, which can also be computationally intensive for large instances. The main goal of the paper is to introduce an efficient and scalable first-order method to address these limitations.

## 3. Methodology

We begin with reformulating (5) as the following *unconstrained* optimization problem

$$\min_{\boldsymbol{\beta}} f(\boldsymbol{X}\boldsymbol{\beta}, \boldsymbol{y}) + 2\lambda_2 \, g(\boldsymbol{\beta}), \quad (6)$$

where the implicit function $g : \mathbb{R}^p \to \mathbb{R} \cup \{\infty\}$ is defined as

$$g(\boldsymbol{\beta}) = \begin{cases} \min_{\boldsymbol{z} \in \mathbb{R}^p} & \frac{1}{2} \sum_{j \in [p]} \beta_j^2 / z_j \\ \text{s.t.} & \boldsymbol{z} \in [0, 1]^p, \, \mathbf{1}^\top \boldsymbol{z} \le k, \\ & -M z_j \le \beta_j \le M z_j \, \forall j \in [p]. \end{cases} \quad (7)$$

Here, we follow the standard convention that an infeasible minimization problem is assigned a value of $+\infty$. Note that $g$ is convex as convexity is preserved under partial minimization over a convex set (Rockafellar, 1970, Theorem 5,3). Furthermore, as $f$ is assumed to be Lipschitz smooth and $g$ is non-smooth, problem (6) is an unconstrained optimization problem with a convex composite objective function. As such, it is amenable to be solved using the FISTA algorithm proposed in (Beck & Teboulle, 2009).

In the following, we first analyze the conjugate of $g$. We then propose an efficient numerical method to compute the proximal operator of $g^*$. This, in turn, enables us to compute the proximal operator of $g$, leading to an efficient implementation of the FISTA algorithm. To further enhance the performance of FISTA, we present an efficient approach to solve the minimization problem (7), which guides us in developing an effective restart procedure. Finally, we conclude this section by providing efficient lower bounds for each step of the BnB framework.

### 3.1. Conjugate function $g^*$

Recall that the conjugate of $g$ is defined as

$$g^*(\boldsymbol{\alpha}) = \sup_{\boldsymbol{\beta} \in \mathbb{R}^p} \boldsymbol{\alpha}^\top \boldsymbol{\beta} - g(\boldsymbol{\beta}).$$

The following lemma gives a closed-form expression for $g^*$, where $\text{TopSum}_k(\cdot)$ denotes the sum of the top $k$ largest elements, and $H_M : \mathbb{R} \to \mathbb{R}$ is the Huber loss function defined as

$$H_M(\alpha_j) := \begin{cases} \frac{1}{2}\alpha_j^2 & \text{if } |\alpha_j| \le M \\ M|\alpha_j| - \frac{1}{2}M^2 & \text{if } |\alpha_j| > M \end{cases}. \quad (8)$$

For notational simplicity, we use the shorthand notation $\mathbf{H}_M(\boldsymbol{\alpha})$ to denote $\mathbf{H}_M(\boldsymbol{\alpha}) = (H_M(\alpha_1), \dots, H_M(\alpha_p))$.

**Lemma 3.1.** *The conjugate of $g$ is given by*

$$g^*(\boldsymbol{\alpha}) = \text{TopSum}_k(\mathbf{H}_M(\boldsymbol{\alpha})). \quad (9)$$

This closed-form expression enables us to compute the proximal of $g^*$. Note that while the proximal operators of both $\text{TopSum}_k$ and $\mathbf{H}_M$ functions are known (see, for example, (Beck, 2017, Examples 6.50 & 6.54)), the conjugate function $g^*$ is defined as the composition of these two functions. However, there is no general formula to derive the proximal operator of a composition of two functions based on the proximal operators of the individual functions. In the next section, we see how to bypass this compositional difficulty.

### 3.2. Proximal operator of $g^*$

Recall that the proximal operator of $g^*$ with weight parameter $\rho > 0$ is defined as

$$\text{prox}_{\rho g^*}(\boldsymbol{\mu}) = \underset{\boldsymbol{\alpha} \in \mathbb{R}^p}{\text{argmin}} \, \frac{1}{2}\|\boldsymbol{\alpha} - \boldsymbol{\mu}\|_2^2 + \rho g^*(\boldsymbol{\alpha}). \quad (10)$$

The evaluation of $\text{prox}_{\rho g^*}$ involves a minimization problem that can be reformulated as a convex SOCP problem. Generic solvers based on IPM and ADMM require solving systems of linear equations. This results in cubic time complexity per iteration, making them computationally expensive, particularly for large-scale problems. These methods also cannot return *exact* solutions. The lack of exactness can affect the stability and reliability of the proximal operator, which is crucial for the convergence of the FISTA algorithm. Inspired by (Busing, 2022), we present Algorithm 1, a customized pooled adjacent violators algorithm that provides an exact evaluation of $\text{prox}_{\rho g^*}$ in linear time after performing a simple 1D sorting step.

**Theorem 3.2.** *For any $\boldsymbol{\mu} \in \mathbb{R}^p$, Algorithm 1 returns the exact evaluation of $\text{prox}_{\rho g^*}(\boldsymbol{\mu})$ in $\mathcal{O}(p \log p)$.*

The proof relies on several auxiliary lemmas. We start with the following lemma, which uncovers a close connection

---

**Algorithm 1** Customized PAVA to solve $\text{prox}_{\rho g^*}(\boldsymbol{\mu})$

---

**Input:** vector $\boldsymbol{\mu}$, scalar multiplier $\rho$, and threshold $M$ of the Huber loss function $H_M(\cdot)$

1: Initialize $\boldsymbol{\rho} \in \mathbb{R}^n$ with $\rho_j = \rho$ if $j \in \{1, 2, ..., k\}$ and $\rho_j = 0$ otherwise.
2: ▷*Sort $\boldsymbol{\mu}$ such that $|\mu_1| \geq |\mu_2| \geq ... \geq |\mu_p|$; $\boldsymbol{\pi}$ is the sorting order.*
3: $\boldsymbol{\mu}, \boldsymbol{\pi} = \text{SpecialSort}(\boldsymbol{\mu})$
4: ▷*STEP 1: Initialize a pool of $p$ blocks with start and end indices; each block initially has length equal to 1*
5: $\mathcal{J} = \{[1, 1], [2, 2], ..., [p, p]\}$
6: ▷*STEP 2: Initialize $\hat{\nu}_j$ in each block by ignoring the isotonic constraints*
7: **for** $j = 1, 2, \ldots, p$ **do**
8:     $\hat{\nu}_j = \text{argmin}_\nu \frac{1}{2}(\nu - |\mu_j|)^2 + \rho_j H_M(\nu)$
9: **end for**
10: ▷*STEP 3: Whenever there is an isotonic constraint violation between two adjacent blocks, merge the two blocks by setting all values to be the minimizer of the objective function restricted to this merged block; use Algorithm 4 in Appendix A*
11: **while** $\exists [a_1, a_2], [a_2 + 1, a_3] \in \mathcal{J}$ s.t. $\hat{\nu}_{a_1} < \hat{\nu}_{a_3}$ **do**
12:     $\mathcal{J} = \mathcal{J} \setminus \{[a_1, a_2]\} \setminus \{[a_2 + 1, a_3]\} \cup \{[a_1, a_3]\}$
13:     $\hat{\nu}_{[a_1:a_3]} = \underset{\nu}{\text{argmin}} \sum_{j=a_1}^{a_3} \left[\frac{1}{2}(\nu - |\mu_j|)^2 + \rho_j H_M(\nu)\right]$
14: **end while**
15: ▷*Return $\hat{\boldsymbol{\nu}}$ with the inverse sorting order*
16: **Return** $\text{sgn}(\boldsymbol{\mu}) \odot \boldsymbol{\pi}^{-1}(\hat{\boldsymbol{\nu}})$

---

between the proximal operator of $g^*$ and the generalized isotonic regression problems.

**Lemma 3.3.** *For any $\boldsymbol{\mu} \in \mathbb{R}^p$, we have*

$$\text{prox}_{\rho g^*}(\boldsymbol{\mu}) = \text{sgn}(\boldsymbol{\mu}) \odot \boldsymbol{\nu}^\star,$$

*where $\odot$ denotes the Hadamard (element-wise) product, $\boldsymbol{\nu}^\star$ is the unique solution of the following optimization problem*

$$\begin{aligned} \min_{\boldsymbol{\nu} \in \mathbb{R}^p} \quad & \frac{1}{2} \sum_{j \in [p]} (\nu_j - |\mu_j|)^2 + \rho \sum_{j \in \mathcal{J}} H_M(\nu_j) \\ \text{s.t.} \quad & \nu_j \geq \nu_l \ \text{if} \ |\mu_j| \geq |\mu_l| \ \forall j, l \in [p], \end{aligned} \quad (11)$$

*and $\mathcal{J}$ is the set of indices of the top $k$ largest elements of $|\mu_j|, j \in [p]$.*

Problem (11) replaces the $\text{TopSum}_k$ in (10) from the conjugate function $g^*$ (as shown in Lemma 3.1) with linear constraints. While this may appear computationally complex, it actually converts the problem into an instance of isotonic regression (Best & Chakravarti, 1990). Such problems can be solved exactly in linear time after performing a simple sorting step. The procedure is known as PAVA (Busing, 2022). Specifically, Algorithm 1 implements a customized

PAVA variant designed to compute $\text{prox}_{\rho g^*}$ exactly. The following lemma shows that the vector generated by Algorithm 1 is an exact solution to (11). Intuitively, Algorithm 1 iteratively merges adjacent blocks until no isotonic constraint violations remain, at which point the resulting vector is guaranteed to be the optimal solution to (11).

**Lemma 3.4.** *The vector $\hat{\boldsymbol{\nu}}$ in Algorithm 1 solves (11) exactly.*

Finally, the merging process in Algorithm 1 can be executed efficiently. Intuitively, each element of $\boldsymbol{\mu}$ is visited at most twice; once during its initial processing and once when it is included in a merged block. This ensures that the process achieves a linear time complexity.

**Lemma 3.5.** *The merging step (lines 11-14) in Algorithm 1 can be performed in linear time complexity $\mathcal{O}(p)$.*

Armed with these lemmas, one can easily prove Theorem 3.2. Details are provided in Appendix A.

### 3.3. FISTA algorithm with restart

A critical computational step in FISTA is the efficient evaluation of the proximal operator of $g$. By the extended Moreau decomposition theorem (Beck, 2017, Theorem 6.45), for any weight parameter $\rho > 0$ and any point $\boldsymbol{\mu} \in \mathbb{R}^p$, the proximal operators of $g$ and $g^*$ satisfies

$$\text{prox}_{\rho^{-1}g}(\boldsymbol{\mu}) = \boldsymbol{\mu} - \rho^{-1} \text{prox}_{\rho g^*}(\rho\boldsymbol{\mu}). \quad (12)$$

Hence, together with Theorem 3.2, we can compute exactly $\text{prox}_{\rho^{-1}g}$ using Algorithm 1 with log-linear time complexity. This enables an efficient implementation of the FISTA algorithm. We note that when $M = \infty$, the implicit function $g$ is closely related to the $k$-support norm. For this special case, Argyriou et al. (2012) and McDonald et al. (2016) have developed efficient algorithms that compute the proximal operator of $g$ in the primal space with log-linear time complexity. Our work extends these results by enabling the incorporation of big-M constraints through a dual analysis.

Besides, the vanilla FISTA algorithm is prone to oscillatory behavior, which results in a sub-linear convergence rate of $\mathcal{O}(1/T^2)$ after $T$ iterations. In the following, we further accelerate the empirical convergence performance of the FISTA algorithm by incorporating a simple restart strategy based on the function value, originally proposed in (O'donoghue & Candes, 2015). In simple terms, the restart strategy operates as follows: if the objective function increases during the iterative process, the momentum coefficient is reset to its initial value. The effectiveness of the restart strategy hinges on the efficient computation of the loss function. This task essentially reduces to evaluating the implicit function $g$ defined in (7), which would involve solving a SOCP problem. However, the value of $g$ can be computed efficiently by leveraging the majorization technique (Kim et al., 2022), as shown in Algorithm 2.

---

**Algorithm 2** Algorithm to compute $g(\boldsymbol{\beta})$

**Input:** vector $\boldsymbol{\beta} \in \mathbb{R}^p$ from Step 2 Line 8 in Algorithm 3.

1: Initialize: $\boldsymbol{\psi} = \mathbf{0} \in \mathbb{R}^k$
2: Sort $\boldsymbol{\beta}$ partially such that
$$|\beta_1| \geq |\beta_2| \geq ... \geq |\beta_k| \geq \max_{k+1,...,p} \{|\beta_j|\}$$
3: $s = \sum_{j=1}^p |\beta_j|$
4: ▷*Compute the majorization vector $\boldsymbol{\psi}$; see Appendix 3.6 for its definition and connection with $\boldsymbol{\beta}$*
5: **for** $j = 1, 2, \ldots, k$ **do**
6:     $\bar{s} = s/(k - j + 1)$
7:     **if** $\bar{s} \geq |\beta_j|$ **then** $\psi_{j:k} = \bar{s}$; break **else** $\psi_j = |\beta_j|$
8:     $s = s - |\beta_j|$
9: **end for**
10: **return** $\frac{1}{2} \sum_{j=1}^k \psi_j^2$

---

**Theorem 3.6.** *For any $\boldsymbol{\beta} \in \mathbb{R}^p$, Algorithm 2 computes the exact value of $g(\boldsymbol{\beta})$, defined in* (7), *in* $\mathcal{O}(p + p \log k)$.

Theorem 3.6 guarantees that Algorithm 2 can efficiently compute the value of $g(\boldsymbol{\beta})$, which is crucial for our value-based restart strategy to be effective in practice. Empirically, we observe that the function value-based restart strategy can accelerate FISTA from the sub-linear convergence rate of $O(1/T^2)$ to a linear convergence rate in many empirical results. To the best of our knowledge, *this is the first linear convergence result of using a first-order method in the MIP context* when calculating the lower bounds in the BnB tree. The FISTA algorithm is summarized in Algorithm 3.

### 3.4. Safe Lower Bounds for GLMs

We conclude this section by commenting on how to use Algorithm 3 in the BnB tree to prune nodes. As an itera-

---

**Algorithm 3** Main algorithm to solve problem (5)

**Input:** number of iterations $T$, coefficient $\lambda_2$ for the $\ell_2$ regularization, and step size $L$ (Lipschitz-continuity parameter of $\nabla F(\boldsymbol{\beta})$)

1: Initialize: $\boldsymbol{\beta}^0 = \mathbf{0}, \boldsymbol{\beta}^1 = \mathbf{0}, \phi = 1$
2: Let: $\rho = L/(2\lambda_2), \mathcal{L}^1 = f(\boldsymbol{\beta}^1, \boldsymbol{y})$
3: **for** $t = 1, 2, 3, ..., T$ **do**
4:     ▷*Step 1: momentum acceleration*
5:     $\boldsymbol{\gamma}^t = \boldsymbol{\beta}^t + \frac{\phi}{\phi+3}(\boldsymbol{\beta}^t - \boldsymbol{\beta}^{t-1})$
6:     ▷*Step 2: proximal gradient descent; use Algorithm 1*
7:     $\boldsymbol{\gamma}^t = \boldsymbol{\gamma}^t - \frac{1}{L}\nabla F(\boldsymbol{\gamma}^t)$
8:     $\boldsymbol{\beta}^{t+1} = \boldsymbol{\gamma}^t - \rho^{-1}\text{prox}_{\rho g^*}(\rho\boldsymbol{\gamma}^t)$
9:     ▷*Step 3: restart; use Algorithm 2*
10:     $\mathcal{L}^{t+1} = f(\boldsymbol{X}\boldsymbol{\beta}^{t+1}, \boldsymbol{y}) + 2\lambda_2 g(\boldsymbol{\beta}^{t+1})$
11:     **if** $\mathcal{L}^{t+1} \geq \mathcal{L}^t$ **then** $\phi = 1$ **else** $\phi = \phi + 1$
12: **end for**
13: **return** $\boldsymbol{\beta}^{T+1}$

---

tive algorithm, FISTA yields only an approximate solution $\hat{\boldsymbol{\beta}}$ to (5). Consequently, while we can calculate the objective function for $\hat{\boldsymbol{\beta}}$ efficiently, this value is not necessarily a lower bound of the original problem–only the optimal value of the relaxed problem (5) serves as a guaranteed lower bound. To get a safe lower bound, we rely on the weak duality theorem, in which for any proper, lower semi-continuous, and convex functions $F : \mathbb{R}^n \to \mathbb{R} \cup \{\infty\}$ and $G : \mathbb{R}^p \to \mathbb{R} \cup \{\infty\}$, we have

$$\inf_{\boldsymbol{\beta} \in \mathbb{R}^p} F(\boldsymbol{X}\boldsymbol{\beta}) + G(\boldsymbol{\beta}) \geq \sup_{\boldsymbol{\zeta} \in \mathbb{R}^n} -F^*(-\boldsymbol{\zeta}) - G^*(\boldsymbol{X}^\top \boldsymbol{\zeta})$$
$$\geq -F^*(-\boldsymbol{\zeta}) - G^*(\boldsymbol{X}^\top \boldsymbol{\zeta}) \ \forall \boldsymbol{\zeta} \in \mathbb{R}^n;$$

where $F^*$ and $G^*$ denote the conjugates of $F$ and $G$, respectively, while the second inequality follows from the definition of the supremum operator. Letting $F(\boldsymbol{X}\boldsymbol{\beta}) = f(\boldsymbol{X}\boldsymbol{\beta}, \boldsymbol{y}), G(\boldsymbol{\beta}) = 2\lambda_2 g(\boldsymbol{\beta})$ and $\boldsymbol{\zeta} = \nabla F(\boldsymbol{X}\hat{\boldsymbol{\beta}})$, where $\hat{\boldsymbol{\beta}}$ is the output of the FISTA Algorithm 3, we arrive at the safe lower bound

$$P_{\text{MIP}}^\star \geq -F^*(-\hat{\boldsymbol{\zeta}}) - G^*(\boldsymbol{X}^\top \hat{\boldsymbol{\zeta}}), \tag{13}$$

where the inequality follows from the relaxation bound $P_{\text{MIP}}^\star \geq P_{\text{conv}}^\star$ and the weak duality theorem. For convenience, we provide a list of $F^*(\cdot)$ for different GLM loss functions in D.1. The readers are also referred to D.2, where we derive the safe lower bound for the linear regression problem with eigen-perspective relaxation as an example.

## 4. Experiments

We evaluate our proposed methods using both synthetic and real-world datasets to address three key empirical questions:

◇ How fast is our customized PAVA algorithm in evaluating $\text{prox}_g$ compared to existing solvers?

◇ How fast is our proposed FISTA method in calculating the lower bounds compared to existing solvers?

◇ How fast is our customized BnB algorithm compared to existing solvers?

We implement our algorithms in python. For baselines, we compare with the following state-of-the-art commercial and open-source SOCP solvers: Gurobi (Gurobi Optimization, LLC, 2025), MOSEK (ApS, 2025), SCS (O'Donoghue et al., 2023), and Clarabel (Goulart & Chen, 2024), with the python package cvxpy (Diamond & Boyd, 2016) as the interface to these solvers.

### 4.1. How Fast Can We Evaluate $\text{prox}_{\rho^{-1}g}(\cdot)$?

In this subsection, we demonstrate the computational efficiency of using our PAVA algorithm for evaluating the proximal operators. We conduct the comparisons in two ways — evaluating both a) the proximal operator of the original function $g$ and b) the proximal operator of its conjugate

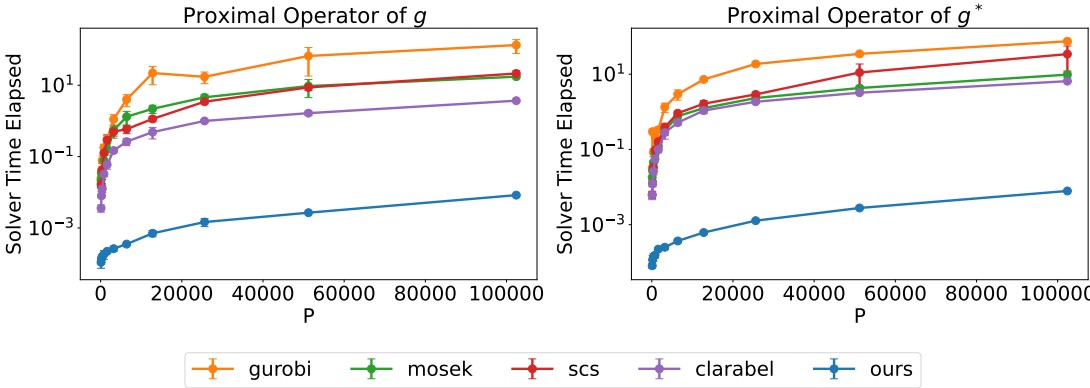

*Figure 1.* Running time comparison of evaluating the proximal operators, for both $g$ (left) and $g^*$ (right). The baselines evaluate the proximal operators by directly solving the corresponding second-order conic problems (SOCP), respectively.

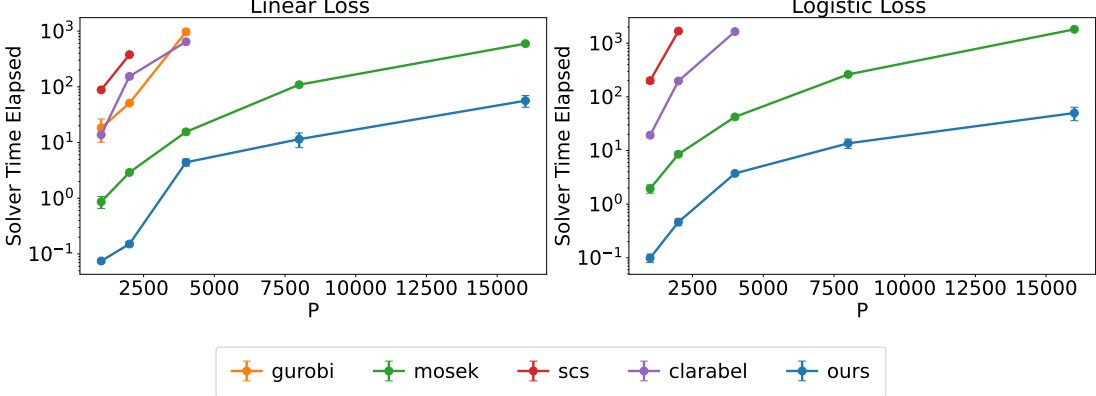

*Figure 2.* Running time comparison of solving Problem (5), the perspective relaxation of the original MIP problem. We set $M = 2.0$, $\lambda_2 = 1.0$, and $n$-to-$p$ ratio to be 1. Gurobi cannot solve the cardinality constrained logistic regression problem.

*Table 1.* GPU acceleration of our method on the linear regression task. Top and bottom rows correspond to the mean and standard deviation of running times (seconds).

| $p$ | 1k | 2k | 4k | 8k | 16k |
|---|---|---|---|---|---|
| ours CPU | 0.19 (0.01) | 0.48 (0.05) | 1.54 (0.21) | 4.80 (0.57) | 19.52 (1.27) |
| ours GPU | 0.29 (0.04) | 0.19 (0.02) | 0.26 (0.02) | 0.59 (0.08) | 2.09 (0.11) |

$g^*$. Detailed experimental configurations, including parameter specifications and synthetic data generation process, are provided in Appendix B.1.

The results shown in Figure 1 highlight the superiority of our method. Our algorithm achieves a computational speedup of approximately two orders of magnitude compared to conventional SOCP solvers. This performance gain is largely due to our customized PAVA implementation in Algorithm 1. For instance, in high-dimensional settings ($p = 10^5$), baseline methods require several seconds to minutes to evaluate the proximal operators, whereas our approach completes the same task in 0.01 seconds. Additionally, our method guarantees *exact* solutions to the optimization problem, in contrast

to the approximate solutions returned by the baselines. This combination of precision and efficiency constitutes a critical advantage for our first-order optimization framework over generic conic programming solvers, as demonstrated in subsequent sections.

### 4.2. How Fast Can We Calculate the Lower Bound?

We next benchmark the computational speed and scalability of our method against the state-of-the-art solvers (Gurobi, MOSEK, SCS, and Clarabel) for solving the perspective relaxation of the original MIP problem. Evaluations are performed on both linear and logistic regression tasks. Experimental configurations are detailed in Appendix B.2. Additional perturbation studies, such as on the sample-to-feature ($n$-to-$p$) ratio, box constraint $M$, and $\ell_2$ regularization coefficient $\lambda_2$, are provided in Appendix C.1. All solvers are terminated upon achieving an optimality gap tolerance of $\epsilon = 10^{-6}$ or exceeding a runtime limit of 1800 seconds.

The results, shown in Figure 2, demonstrates that our method outperforms the fastest conic solver (MOSEK) by over one order of magnitude. For the largest tested instances ($n = 16000$ and $p = 16000$), our approach attains the target

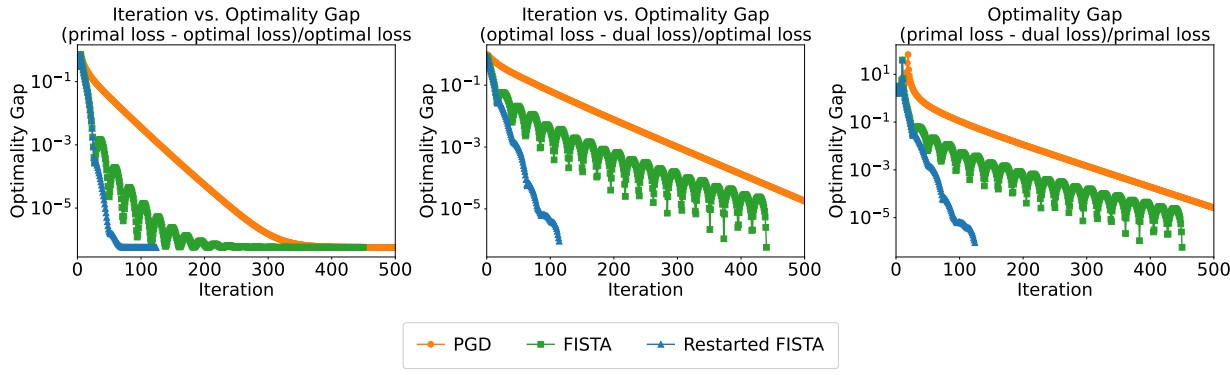

*Figure 3.* Empirical convergence rate of our restarted FISTA (compared with PGD, the proximal gradient method, and FISTA) on solving the perspective relaxation in Problem (5) with the logistic loss, $n = 16000, p = 16000, k = 10, \rho = 0.5, \lambda_2 = 1.0,$ and $M = 2.0$.

*Table 2.* Certifying optimality on large-scale and real-world datasets.

| | | ours | | Gurobi | | MOSEK | |
|---|---|---|---|---|---|---|---|
| | | time (s) | opt. gap (%) | time (s) | opt. gap (%) | time (s) | opt. gap (%) |
| Linear Regression | synthetic ($k = 10, M = 2$) (n=16k, p=16k, seed=0) | 79 | 0.0 | 1800 | - | 1915 | - |
| | Cancer Drug Response ($k = 5, M = 5$) (n=822, p=2300) | 41 | 0.0 | 1800 | 0.89 | 188 | 0.0 |
| Logistic Regression | Synthetic ($k = 10, M = 2$) (n=16k, p=16k, seed=0) | 626 | 0.0 | N/A | N/A | 2446 | - |
| | DOROTHEA ($k = 10, M = 2$) (n=2300, p=89989) | 230 | 0.0 | N/A | N/A | 1814 | 0.63 |

tolerance ($10^{-6}$) in under 100 seconds across regression and classification datasets, whereas most baselines fail to converge within the 1800-second threshold.

There are two factors driving this speedup. First, our efficient proximal operator evaluation reduces per-iteration complexity. Second, our efficient method to compute $g(\boldsymbol{\beta})$ (in Algorithm 2) exactly enables integration of the value-based restart technique within FISTA, significantly improving convergence. Figure 3 illustrate this enhancement: while PGD (the proximal gradient descent method, also known as ISTA) and FISTA exhibit slow convergence rates, FISTA with restarts achieves fast linear convergence on both dual loss and primal-dual gap metrics. To the best of our knowledge, this marks the first empirical demonstration of linear convergence for a first-order method applied to solving the convex relaxation of this MIP class. Finally, our method permits GPU acceleration because our most computationally intensive component is matrix-vector multiplications. As shown in Table 1, GPU implementation reduces runtime by an additional order of magnitude on high-dimensional instances.

### 4.3. How Fast Can We Certify Optimality?

Finally, we demonstrate how our method's ability to compute tight lower bounds enables efficient optimality certification for large-scale datasets, outperforming state-of-the-art

commerical MIP solvers. Integrating our lower-bound computation into a minimalist branch-and-bound (BnB) framework, we prioritize node pruning via lower bound calculations while intentionally omitting advanced MIP heuristics (e.g., cutting planes, presolve routines) to evaluate the impact of our method. Experimental configurations, including dataset descriptions and BnB implementation details, are provided in Appendix B.3. We benchmark our approach against Gurobi and MOSEK, reporting both runtime and final optimality gaps. Note that on the two real-world datasets, we have used small $k$'s, which are selected based on 5-fold cross validation (see Appendix B.3). Small $k$'s are sufficient for accurate prediction and can help avoid overfitting. They also improve interpretability of the model.

Results in Table 2 show that our method certifies optimality for 2 of the four tested datasets around 1 minute, the third around 10 minutes, and the fourth around 4 minutes. In contrast, Gurobi and MOSEK either exceed the time limit (1800 seconds) during the presolve stage or require significantly longer runtimes to achieve zero or small gaps. Crucially, this efficiency stems from our efficient lower-bound computations and dynamic early termination criteria. Specifically, we avoid waiting for full convergence by leveraging two key rules: (1) if the primal loss falls below the incumbent solution's loss, we terminate early and proceed to branching; (2) if the dual loss exceeds the incumbent's loss, we halt

computation and prune the node immediately. This adaptive approach eliminates unnecessary iterations while ensuring we prune the search space effectively.

## 5. Conclusion

We introduce a first-order proximal algorithm to solve the perspective relaxation of cardinality-constrained GLM problems. By leveraging the problem's unique mathematical structure, we design a customized PAVA to efficiently evaluate the proximal operator, ensuring scalability to high-dimensional settings. Further acceleration is achieved through an efficient value-based restart strategy and compatibility with GPUs, which collectively enhance convergence rates and computational speed. Extensive empirical results demonstrate that our method outperforms state-of-the-art solvers by 1-2 orders of magnitude, establishing it as a practical, high-performance component for integration into next-generation MIP solvers.

## Code Availability

Implementations discussed in this paper are available at https://github.com/jiachangliu/OKGLM.

## Acknowledgments

This work used the Delta system at the National Center for Supercomputing Applications through allocation CIS250029 from the Advanced Cyberinfrastructure Coordination Ecosystem: Services & Support (ACCESS) program, which is supported by National Science Foundation grants #2138259, #2138286, #2138307, #2137603, and #2138296.

## Impact Statement

This paper presents work whose goal is to advance the field of Machine Learning. There are many potential societal consequences of our work, none which we feel must be specifically highlighted here.

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

# Appendix

## Appendix A. Proofs

This section contains all omitted proofs in the paper.

### A.1. Proof of Lemma 2.1

**Lemma 2.1.** *The closed convex hull of the set*

$$\left\{(\tau, \boldsymbol{\beta}, \boldsymbol{z}) \middle| \|\boldsymbol{\beta}\|_\infty \le M,\ \boldsymbol{z} \in \{0,1\}^p,\ \mathbf{1}^\top \boldsymbol{z} \le k,\ \beta_j(1 - z_j) = 0\ \forall j \in [p],\ \sum_{j \in [p]} \beta_j^2 \le \tau\right\}$$

*is given by the set*

$$\left\{(\tau, \boldsymbol{\beta}, \boldsymbol{z})\ \middle|\ -M z_j \le \beta_j \le M z_j\ \forall j \in [p],\ \boldsymbol{z} \in [0,1]^p,\ \mathbf{1}^\top \boldsymbol{z} \le k,\ \sum_{j \in [p]} \beta_j^2 / z_j \le \tau\right\}.$$

*Proof.* Let $\mathcal{T}$ represent the first set mentioned in the statement of the lemma. Using the definition of the perspective function and applying the big-M formulation technique, we have

$$\mathcal{T} = \left\{(\tau, \boldsymbol{\beta}, \boldsymbol{z})\ \middle|\ -M z_j \le \beta_j \le M z_j\ \forall j \in [p],\ \boldsymbol{z} \in \{0,1\}^p,\ \mathbf{1}^\top \boldsymbol{z} \le k,\ \sum_{j \in [p]} \beta_j^2 / z_j \le \tau\right\}.$$

As the epigraph of a perspective function constitutes a cone (Shafiee & Kılınç-Karzan, 2024, Lemma 1 & 2), we may write $\mathcal{T} = \mathrm{Proj}_{(\tau, \boldsymbol{\beta}, \boldsymbol{z})}(\overline{\mathcal{T}})$, where

$$\overline{\mathcal{T}} = \left\{(\tau, \boldsymbol{\beta}, \boldsymbol{t}, \boldsymbol{z})\ \middle|\ \mathbf{1}^\top \boldsymbol{t} = \tau,\ \boldsymbol{z} \in \{0,1\}^p,\ \mathbf{1}^\top \boldsymbol{z} \le k,\ \boldsymbol{A}_j \begin{bmatrix} t_j \\ \beta_j \end{bmatrix} + \boldsymbol{B}_j z_j \in \mathbb{K}_j\ \forall j \in [p]\right\}$$

admits a mixed-binary conic representation with

$$\boldsymbol{A} = \begin{bmatrix} 1 & 0 \\ 0 & 1 \\ 0 & 0 \\ 0 & 1 \\ 0 & -1 \end{bmatrix},\ \boldsymbol{B} = \begin{bmatrix} 0 \\ 0 \\ 0 \\ M \\ M \end{bmatrix},\ \mathbb{K}_j = \mathbb{L}_+ \times \mathbb{R}_+ \times \mathbb{R}_+ \qquad \forall j \in [p].$$

Here, $\mathbb{L}_+ \in \mathbb{R}^3$ denotes the rotated second order cone, that is, $\mathbb{L}_+ = \{(t, \beta, z) \in \mathbb{R}_+ \times \mathbb{R} \times \mathbb{R}_+ : \beta^2 \le tz\}$. Thus, using (Shafiee & Kılınç-Karzan, 2024, Lemma 4), the set $\overline{\mathcal{T}}$ satisfies all the requirements of (Shafiee & Kılınç-Karzan, 2024, Theorem 1), and therefore, its continuous relaxation gives the closed convex hull of $\overline{\mathcal{T}}$, that is,

$$\mathrm{cl\,conv}(\overline{\mathcal{T}}) = \left\{(\tau, \boldsymbol{\beta}, \boldsymbol{t}, \boldsymbol{z})\ \middle|\ \mathbf{1}^\top \boldsymbol{t} = \tau,\ \boldsymbol{z} \in [0,1]^p,\ \mathbf{1}^\top \boldsymbol{z} \le k,\ \boldsymbol{A}_j \begin{bmatrix} t_j \\ \beta_j \end{bmatrix} + \boldsymbol{B}_j z_j \in \mathbb{K}_j\ \forall j \in [p]\right\}.$$

The prove concludes by applying Fourier-Motzkin elimination method to project out the variable $\boldsymbol{t}$. $\qquad \square$

### A.2. Proof of Lemma 3.1

**Lemma 3.1.** *The conjugate of $g$ is given by*

$$g^*(\boldsymbol{\alpha}) = \mathrm{TopSum}_k(\mathbf{H}_M(\boldsymbol{\alpha})).$$

*Proof.* Using the definition of the implicit function $g$ in (7), we have

$$g^*(\boldsymbol{\alpha}) = \begin{cases} \max & \boldsymbol{\alpha}^\top \boldsymbol{\beta} - \frac{1}{2} \sum_{j \in [p]} \beta_j^2 / z_j \\ \text{s.t.} & \boldsymbol{\beta} \in \mathbb{R}^p,\ \boldsymbol{z} \in [0,1]^p,\ \mathbf{1}^\top \boldsymbol{z} \le k, \\ & -M z_j \le \beta_j \le M z_j\ \forall j \in [p] \end{cases} \tag{14}$$

For any fixed feasible $\boldsymbol{z}$, the maximization problem over $\boldsymbol{\beta}$ is a simple constrained quadratic problem, that can be solved analytically by the vector $\boldsymbol{\beta}^\star$ whose $j$'th element is given by $\beta_j^\star = \mathrm{sgn}(\alpha_j)\min(|\alpha_j|, M)z_j$. Substituting the optimizer $\boldsymbol{\beta}^\star$, the objective function of the maximization problem in (14) simplifies to

$$\boldsymbol{\alpha}^\top \boldsymbol{\beta}^\star - \frac{1}{2}\sum_{j\in[p]} \beta_j^{\star 2}/z_j = \sum_{j\in[p]} \alpha_j \cdot \mathrm{sgn}(\alpha_j)\min(|\alpha_j|, M)z_j - \frac{(\mathrm{sgn}(\alpha_j)\min(|\alpha_j|, M)z_j)^2}{2z_j}$$

$$= \sum_{j\in[p]} \left( |\alpha_j|\min(|\alpha_j|, M) - \frac{1}{2}\min(\alpha_j^2, M^2) \right) z_j$$

$$= \sum_{j\in[p]} H_M(\alpha_j)z_j,$$

where the second equality holds as $\boldsymbol{z}$ is a binary vector, and the last equality follows from the definition of the Huber loss function:

$$H_M(x) = \begin{cases} \frac{1}{2}x^2 & \text{if } |x| \leq M \\ M|x| - \frac{1}{2}M^2 & \text{if } |x| > M \end{cases}.$$

We thus arrive at

$$g^*(\boldsymbol{\alpha}) = \max_{\boldsymbol{z}\in[0,1]^p} \left\{ \sum_{j\in[p]} H_M(\alpha_j)z_j : \mathbf{1}^\top \boldsymbol{z} \leq k \right\} = \mathrm{TopSum}_k(\mathbf{H}_M(\boldsymbol{\alpha})).$$

This completes the proof.

$\square$

Note that recent works (Xie et al., 2020; Lei et al., 2023) have discussed the smooth approximation of the $\mathrm{TopSum}_k(\cdot)$. However, such a smooth approximation is not suitable for this work. The effectiveness of the proximal algorithm relies on the exact evaluation of the proximal operator, which we will talk about next. Replacing $\mathrm{TopSum}_k(\cdot)$ would lead to solving a different problem rather than the proximal operator evaluation. This will not help us to use FISTA to solve the perspective relaxation. As a result, this approach would not guarantee valid lower bounds necessary for optimality certification.

### A.3. Proof of Lemma 3.3

**Lemma 3.3.** *For any $\boldsymbol{\mu} \in \mathbb{R}^p$, we have*

$$\mathrm{prox}_{\rho g^*}(\boldsymbol{\mu}) = \mathrm{sgn}(\boldsymbol{\mu}) \odot \boldsymbol{\nu}^\star,$$

*where $\odot$ denotes the Hadamard (element-wise) product, $\boldsymbol{\nu}^\star$ is the unique solution of the following optimization problem*

$$\min_{\boldsymbol{\nu}\in\mathbb{R}^p} \quad \frac{1}{2}\sum_{j\in[p]}(\nu_j - |\mu_j|)^2 + \rho\sum_{j\in\mathcal{J}} H_M(\nu_j) \tag{15}$$

$$\text{s.t.} \quad \nu_j \geq \nu_l \ \text{if } |\mu_j| \geq |\mu_l| \ \forall j, l \in [p],$$

*and $\mathcal{J}$ is the set of indices of the top $k$ largest elements of $|\mu_j|, j \in [p]$.*

*Proof.* For simplicity, let $\boldsymbol{\alpha}^\star = \mathrm{prox}_{\rho g^*}(\boldsymbol{\mu})$, that is,

$$\boldsymbol{\alpha}^\star = \underset{\boldsymbol{\alpha}\in\mathbb{R}^p}{\mathrm{argmin}} \ \frac{1}{2}\|\boldsymbol{\alpha} - \boldsymbol{\mu}\|_2^2 + \rho g^*(\boldsymbol{\alpha}). \tag{16}$$

We first show that $\mathrm{sgn}(\boldsymbol{\alpha}^\star) = \mathrm{sgn}(\boldsymbol{\mu})$ (step 1) and then establish that for every $j, l \in [p]$ with $|\mu_j| \geq |\mu_l|$, we have $|\alpha_j^\star| \geq |\alpha_l^\star|$ (step 2). We then conclude the proof using these observations.

◇ **Step 1.** We prove the sign-preserving property through a proof by contradiction. For the sake of contradiction, suppose that there exists some $j \in [p]$ such that $\mathrm{sgn}(\alpha_j^\star) \neq \mathrm{sgn}(\mu_j)$. Hence, we can construct a new $\boldsymbol{\alpha}'$ by flipping the sign of $\alpha_j^\star$, i.e., $\alpha_j' = -\alpha_j^\star$, and keeping the rest of the elements the same as $\boldsymbol{\alpha}^\star$. Now under the assumption that $\mathrm{sgn}(\alpha_j^\star) \neq \mathrm{sgn}(\mu_j)$, we have $|\alpha_j^\star - \mu_j| > ||\alpha_j^\star| - |\mu_j|| = |\alpha_j' - \mu_j|$, so the $j$-th term in the first summation of the objective function will decrease while everything else remains the same. This leads to a smaller objective value for $\boldsymbol{\alpha}'$ than $\boldsymbol{\alpha}^\star$, which contradicts the optimality of $\boldsymbol{\alpha}^\star$. Thus, the claim follows.

⬦ **Step 2.** We prove the relative magnitude-preserving property through a proof by contradiction. For the sake of contradiction, suppose that there exists some $j, l \in [p]$ such that $|\mu_j| \geq |\mu_l|$ but $|\alpha_j^\star| < |\alpha_l^\star|$. Then, we can construct a new $\boldsymbol{\alpha}'$ by swapping $\alpha_j^\star$ and $\alpha_l^\star$, i.e., $\alpha_j' = \alpha_l^\star$ and $\alpha_l' = \alpha_j^\star$, and keeping the rest of the elements the same as $\boldsymbol{\alpha}^\star$. Under the assumption that $|\mu_j| \geq |\mu_l|$ but $|\alpha_j^\star| < |\alpha_l^\star|$, we have $\left|\alpha_j^\star - \mu_j\right| + \left|\alpha_l^\star - \mu_l\right| > \left|\alpha_l^\star - \mu_j\right| + \left|\alpha_j^\star - \mu_l\right| = \left|\alpha_j' - \mu_j\right| + \left|\alpha_l' - \mu_l\right|$, so the sum of the $j$-th and $l$-th terms in the first summation of the objective function will decrease while everything else remains the same. This leads to a smaller objective value for $\boldsymbol{\alpha}'$ than $\boldsymbol{\alpha}^\star$, which contradicts the optimality of $\boldsymbol{\alpha}^\star$. Thus, the claim follows.

Using these two observations, we are ready to prove that $\boldsymbol{\alpha}^\star = \mathrm{sgn}(\boldsymbol{\mu}) \odot \boldsymbol{\nu}^\star$. We first reparametrize the minimization problem (16) by substituting the decision variable $\boldsymbol{\alpha}$ with a new variable $\boldsymbol{\nu} \in \mathbb{R}_+^p$ satisfying $\boldsymbol{\alpha} = \mathrm{sgn}(\boldsymbol{\mu}) \odot \boldsymbol{\nu}$. By the sign-preserving property in step 1, it is easy to show the equivalence between the optimization problem in (16) and the following optimization problem

$$\min_{\boldsymbol{\nu} \in \mathbb{R}_+^p} \ \tfrac{1}{2} \textstyle\sum_{j \in [p]} (\nu_j - |\mu_j|)^2 + \rho \, \mathrm{TopSum}_k \left( \mathbf{H}_M(\boldsymbol{\nu}) \right).$$

By the relative magnitude-preserving property in step 2, we can further set the equivalence between the minimization problem in (16) and the following optimization problem

$$\begin{aligned}
\min_{\boldsymbol{\nu} \in \mathbb{R}_+^p} \quad & \tfrac{1}{2} \textstyle\sum_{j \in [p]} (\nu_j - |\mu_j|)^2 + \rho \textstyle\sum_{j \in \mathcal{J}} H_M(\nu_j), \\
\text{s.t.} \quad & \nu_j \geq \nu_l \ \text{if} \ |\mu_j| \geq |\mu_l|.
\end{aligned}$$

Lastly, the nonnegative constraint on $\boldsymbol{\nu}$ can be removed as the second summation term in the objective function implies that $\nu_j \geq 0$. Thus, we have shown that any feasible point $\boldsymbol{\alpha}$ in the minimization problem (16) can be reconstructed by any feasible point $\boldsymbol{\nu}$ in the minimization problem in the statement of lemma, while maintaining the same objective value. Hence, we may conclude that $\boldsymbol{\alpha}^\star = \mathrm{sgn}(\boldsymbol{\mu}) \odot \boldsymbol{\nu}^\star$, as required. $\qquad\square$

### A.4. Proof of Lemma 3.4

**Lemma 3.4.** *The vector $\hat{\boldsymbol{\nu}}$ in Algorithm 1 solves (11) exactly.*

*Proof.* The minimization problem (11) is an instance of a generalized isotonic regression problem taking the form

$$\min_{\boldsymbol{\nu}} \sum_{j=1}^{p} h_j(\nu_j) \quad \text{s.t.} \quad \nu_1 \geq \nu_2 \geq \cdots \geq \nu_J, \tag{17}$$

where $h_j(\nu) = \frac{1}{2}(\nu - \mu_j)^2 + \rho_j H_M(\nu)$, $\rho_j = \rho$ if $j \in \mathcal{J}$ and $\rho_j = 0$ otherwise, and the set $\mathcal{J}$ is the set of indices of top k largest elements of $|\mu_j|$, as defined in the statement of Lemma 3.3. Thanks to (Best et al., 2000; Ahuja & Orlin, 2001), the optimizer of (17) satisfies two key properties:

⬦ **Property 1: Optimal solution for a merged block is single-valued.** Suppose we have two adjacent blocks $[a_1, a_2]$ and $[a_2 + 1, a_3]$ such that the optimal solution of each block is single-valued, that is, the minimization problems

$$\begin{cases}
\min\limits_{\boldsymbol{\nu}_{a_1 : a_2}} & \sum_{j=a_1}^{a_2} h_j(\nu_j) \\
\text{s.t.} & \nu_{a_1} \geq \cdots \geq \nu_{a_2}
\end{cases} \quad \text{and} \quad \begin{cases}
\min\limits_{\boldsymbol{\nu}_{a_2+1 : a_3}} & \sum_{j=a_2+1}^{a_3} h_j(\nu_j) \\
\text{s.t.} & \nu_{a_2+1} \geq \cdots \geq \nu_{a_3}
\end{cases}$$

are solved by $\boldsymbol{\nu}_{a_1 : a_2}^\star$ and $\boldsymbol{\nu}_{a_2+1 : a_3}^\star$ with $\nu_{a_1}^\star = \cdots = \nu_{a_2}^\star$ and $\nu_{a_2+1}^\star = \cdots = \nu_{a_3}^\star$, respectively. If $\nu_{a_1}^\star \leq \nu_{a_2+1}^\star$, then the optimal solution for the merged block $[a_1, a_3]$ is single-valued, that is, the minimization problem

$$\begin{cases}
\min\limits_{\boldsymbol{\nu}_{a_1 : a_3}} & \sum_{j=a_1}^{a_3} h_j(\nu_j) \\
\text{s.t.} & \nu_{a_1} \geq \cdots \geq \nu_{a_3}
\end{cases}$$

is solved by $\boldsymbol{\nu}_{a_1 : a_3}^\star$ with $\nu_{a_1}^\star = \cdots = \nu_{a_3}^\star$.

◇ **Property 2: No isotonic constraint violation between single-valued blocks implies the solution is optimal.** Suppose that we have $s$ blocks $[a_1, a_2], [a_2 + 1, a_3], \ldots, [a_s + 1, a_{s+1}]$ (with $a_1 = 1$ and $a_{s+1} = p$) such that the optimal solution for each block is single-valued, that is, $\nu^\star_{a_l+1} = \cdots = \nu^\star_{a_{l+1}}$ for all $l \in [s]$. Then, if $\hat{\nu}_{a_1} \geq \hat{\nu}_{a_2+1} \geq \ldots \hat{\nu}_{a_s}$, then $\hat{\boldsymbol{\nu}}$ is the optimal solution to (17).

Using these two properties, it is now easy to see why Algorithm 1 returns the optimal solution. We start by constructing blocks which have length 1. The initial value restricted to each block is optimal. Then, we iteratively merge adjacent blocks and update the values of $\nu_j$'s whenever there is a violation of the isotonic constraint. By the first property, the optimal solution for the merged block is single-valued. Therefore, we can compute the optimal solution for the merged block by solving a univariate optimization problem. We keep merging blocks until there is no isotonic constraint violation. When this happens, by construction, the solution for each block is single-valued and optimal. By the second property, the final vector $\hat{\boldsymbol{\nu}}$ is the optimal solution to (17), as required. □

### A.5. Proof of Lemma 3.5

**Lemma 3.5.** *The merging step (lines 11-14) in Algorithm 1 can be performed in linear time complexity $\mathcal{O}(p)$.*

*Proof.* A detailed implementation of line 11-14 (Step 3) of the PAVA Algorithm 1 that achieves a linear time complexity is presented in Algorithm 4. In the following, we first show that Algorithm 4 accomplishes the objective in lines 11-14 of Algorithm 1. We then establish that Algorithm 4 runs in linear time complexity.

To prove the first claim, we show that the parameters $P_b$, $S_b$, and $\nu_b$ amount to

$$P_b = \sum_{j \in \mathcal{B}(b)} \rho_j, \; S_b = \sum_{j \in \mathcal{B}(b)} |\mu_j|, \; \nu_b = \mathrm{prox}_{\sum_{j \in \mathcal{B}(b)} \rho_j H_M}(|\mu_j|)$$

for each block index $b$, where $\mathcal{B}(b)$ denoting the set of indices in the $b$'th block. It is easy to verify that Algorithm 4 recursively computes $P_b$ and $S_b$. Thus, we will focus on $\nu_b$. Note that the computation of the proximal operator in $\nu_b$ is reduced to solving a univariate optimization problem for each $b$ and satisfies

$$\begin{aligned}
\nu_b &= \operatorname*{argmin}_{v \in \mathbb{R}} \sum_{j \in \mathcal{B}(b)} \left( \frac{1}{2}(v - |\mu_j|)^2 + \rho_j H_M(v) \right) \\
&= \operatorname*{argmin}_v \sum_{j \in \mathcal{B}(b)} \left( \frac{1}{2}v^2 - v|\mu_j| + \rho_j H_M(v) \right) \\
&= \operatorname*{argmin}_v \left( \frac{1}{2}v^2 - \frac{S_b}{N_b}|\mu_j| + \frac{P_b}{N_b}H_M(v) \right) = \operatorname*{argmin}_v \left( \frac{1}{2}\left(v - \frac{S_b}{N_b}\right)^2 + \frac{P_b}{N_b}H_M(v) \right) = \mathrm{prox}_{\frac{P_b}{N_b}H_M}\left(\frac{S_b}{N_b}\right).
\end{aligned}$$

Thus, Algorithm 4 merges two adjacent blocks if the isotonic violation persists, and the output of the proximal operator is the minimizer of the univariate function in the merged block. This is exactly the same as the objective in lines 11-14 of Algorithm 1. Hence, the first claim follows.

To show that the algorithm runs in linear time, notice that in the while loop $j \leq p$ in Algorithm 4, the variable $j$ is incremented by 1 in each iteration, and the loop terminates when $j = p$. Although there are two while loops inside the main while loop, the total number of iterations in the two inner while loops is at most $p$. This is because we start with $p$ blocks, and each iteration of the inner while loops either merges two blocks forward or merges two blocks backward. The total number of merging operations is at most $p - 1$. Thus, the total number of iterations in the while loop $j \leq p$ is at most $p$. Lastly, since we can evaluate the proximal operator of the Huber loss function, $\mathbf{H}_M$, in constant time complexity, the total time complexity of Algorithm 4 is $O(p)$. □

### A.6. Proof of Theorem 3.2

**Theorem 3.2.** *For any $\boldsymbol{\mu} \in \mathbb{R}^p$, Algorithm 1 returns the exact evaluation of $\mathrm{prox}_{\rho g^*}(\boldsymbol{\mu})$ in $\mathcal{O}(p \log p)$.*

*Proof.* By Lemmas 3.3 and 3.4, the output of Algorithm 1 computes $\mathrm{prox}_{\rho g^*}$ exactly. The log-linear time complexity statement also holds thanks to Lemma 3.5 and the initial sorting step. □

---

**Algorithm 4** Up and Down Block Algorithm for Merging in PAVA

---

**Input:** vector $\boldsymbol{\mu} \in \mathbb{R}^p$, nonnegative weights $\boldsymbol{\rho} \in \mathbb{R}_+^p$ ($\rho_{[1:k]} = \rho$, $\rho_{k+1:p} = 0$), vector $\hat{\boldsymbol{\nu}}$ ($\hat{\nu}_j = \text{prox}_{\rho_j H_M}(|\mu_j|)$), integer $k \in \mathbb{N}$ (first $k$ elements subject to Huber penalty), and threshold $M > 0$ for the Huber loss function.

---

1: ▷*Initialization for the first block*
2: Initialize $b = 1$, $P_1 = \rho_1$, $S_1 = |\mu_1|$, $N_b = 1$, $\nu_1$, $r_1 = 1$.
3: $\nu_{\text{prev}} = \hat{\nu}_1, j = 2$
4: **while** $j \leq n$ **do**
5: $\quad b = b + 1$
6: $\quad P_b = \rho_j$, $S_b = |\mu_j|$, $N_b = 1$, $\nu = \hat{\nu}_j$
7: $\quad$ ▷*If the value for the current singleton block is greater that of the previous block (isotonic violation), merge the current block with the previous block*
8: $\quad$ **if** $\nu > v_{\text{prev}}$ **then**
9: $\qquad b = b - 1$
10: $\qquad P_b = P_b + \rho_j$, $\ S_b = S_b + |\mu_j|$, $\ N_b = N_b + 1$, $\ \nu = \text{prox}_{\frac{P_b}{N_b} H_M}(\frac{S_b}{N_b})$
11: $\qquad$ ▷*Look forward: keep merging the current block with the next block if the isotonic violation persists*
12: $\qquad$ **while** $j < n$ **and** $\nu \leq \hat{\nu}_j$ **do**
13: $\qquad\quad j = j + 1$
14: $\qquad\quad P_b = P_b + \rho_j$, $\ S_b = S_b + |\mu_j|$, $\ N_b = N_b + 1$, $\ \nu = \text{prox}_{\frac{P_b}{N_b} H_M}(\frac{S_b}{N_b})$
15: $\qquad$ **end while**
16: $\qquad$ ▷*Look backward: keep merging the current block with the previous block if the isotonic violation persists*
17: $\qquad$ **while** $b > 1$ **and** $\nu_{b-1} < \nu$ **do**
18: $\qquad\quad b = b - 1$
19: $\qquad\quad P_b = P_b + P_{b+1}$, $\ S_b = S_b + S_{b+1}$, $\ N_b = N_b + N_{b+1}$, $\ \nu = \text{prox}_{\frac{P_b}{N_b} H_M}(\frac{S_b}{N_b})$
20: $\qquad$ **end while**
21: $\quad$ **end if**
22: $\quad$ ▷*Save the current block's value and the index of the last element in the block*
23: $\quad \nu_b = \nu, r_b = j$
24: $\quad$ ▷*Start fresh on the next element*
25: $\quad \nu_{\text{prev}} = \nu, j = j + 1$
26: **end while**
27: ▷*Modify the output vector to have the same new value for all elements in each block*
28: **for** $l = 1, ..., b$ **do**
29: $\quad \hat{\nu}_{[r_{l-1}+1:r_l]} = \nu_l$
30: **end for**
31: **return** $\hat{\boldsymbol{\nu}}$

---

## A.7. Proof of Theorem 3.6

**Theorem 3.6.** *For any $\boldsymbol{\beta} \in \mathbb{R}^p$, Algorithm 2 computes the exact value of $g(\boldsymbol{\beta})$, defined in (7), in $\mathcal{O}(p + p \log k)$.*

*Proof.* We first show that the algorithm correctly computes the value of $g(\boldsymbol{\beta})$ and then analyze its computational complexity. Define the mixed-binary set

$$\mathcal{S}_0 = \left\{(t, \boldsymbol{\beta}) \;\middle|\; \tfrac{1}{2} \sum_{j \in [p]} \beta_j^2 \leq t, \ \|\boldsymbol{\beta}\|_\infty \leq M, \ \|\boldsymbol{\beta}\|_0 \leq k \right\}.$$

Using the perspective and big-M reformulation techniques, the set $\mathcal{S}_0$ admits the equivalent representation

$$\mathcal{S}_0 = \left\{(t, \boldsymbol{\beta}) \;\middle|\; \exists \boldsymbol{z} \in \{0,1\}^p \text{ s.t. } \tfrac{1}{2} \sum_{j \in [p]} \beta_j^2/z_j \leq t, \ \mathbf{1}^\top \boldsymbol{z} \leq k, \ -M z_j \leq \beta_j \leq M z_j \ \forall j \in [p] \right\}.$$

Following the proof of Lemma 2.1, one can show that the closed convex hull of $\mathcal{S}_0$ is given by

$$\text{cl conv}(\mathcal{S}_0) = \left\{(t, \boldsymbol{\beta}) \;\middle|\; \exists \boldsymbol{z} \in [0,1]^p \text{ s.t. } \tfrac{1}{2} \sum_{j \in [p]} \beta_j^2/z_j \leq t, \ \mathbf{1}^\top \boldsymbol{z} \leq k, \ -M z_j \leq \beta_j \leq M z_j \ \forall j \in [p] \right\}.$$

Therefore, the implicit function $g$ can be written as the evaluation of the support function of $\operatorname{cl}\operatorname{conv}(\mathcal{S}_0)$ at $(1, \mathbf{0})$, that is,

$$g(\boldsymbol{\beta}) = \min\{t : (t, \boldsymbol{\beta}) \in \operatorname{cl}\operatorname{conv}(\mathcal{S}_0)\}. \tag{18}$$

Notice that the set $\mathcal{S}_0$ is sign- and permutation-invariants. Hence, by (Kim et al., 2022, Theorem 4), its closed convex hull admits the following (different) lifted represenation

$$\operatorname{cl}\operatorname{conv}(\mathcal{S}_0) = \left\{ (t, \boldsymbol{\beta}) \;\middle|\; \exists \boldsymbol{\phi} \in \mathbb{R}^p \text{ s.t. } \begin{array}{l} \frac{1}{2} \sum_{j \in [p]} \phi_j^2 \le t,\ |\boldsymbol{\beta}| \preceq_m \boldsymbol{\phi}, \\ 0 \le \phi_k \le \ldots \le \phi_1 \le M, \\ \phi_{k+1} = \phi_{k+2} = \ldots = \phi_n = 0 \end{array} \right\}, \tag{19}$$

where the absolute value operator $|\cdot|$ is applied to a vector in an element-wise fashion, and the constraint $|\boldsymbol{\beta}| \preceq_m \boldsymbol{\phi}$ denotes that $\boldsymbol{\phi}$ majorizes $|\boldsymbol{\beta}|$, that is,

$$|\boldsymbol{\beta}| \preceq_m \boldsymbol{\phi} \quad \Longleftrightarrow \quad \sum_{j \in [l]} |\beta_j| \le \sum_{j \in [l]} \phi_j \quad \forall l \in [p-1] \quad \text{and} \quad \sum_{j \in [p]} \phi_j = \sum_{j \in [p]} |\beta_j|.$$

Using this alternative convex hull description of $\mathcal{S}_0$ in (19) and the implicit formulation (18), we may conclude that

$$g(\boldsymbol{\beta}) = \min_{\boldsymbol{\phi} \in \mathbb{R}^p} \left\{ \tfrac{1}{2} \sum_{j \in [p]} \phi_j^2 : |\boldsymbol{\beta}| \preceq_m \boldsymbol{\phi}, 0 \le \phi_k \le \ldots \le \phi_1 \le M, \phi_{k+1} = \phi_{k+2} = \ldots = \phi_n = 0 \right\}. \tag{20}$$

In the following we show that Algorithm 2 can efficiently solve the minimization problem in (20). At the first iteration $j = 1$ of the algorithm, we have

$$k\phi_1 \ge \sum_{j \in [k]} \phi_j = \sum_{j \in [p]} \phi_j \ge \sum_{j \in [p]} |\beta_j| \quad \Rightarrow \quad \phi_1 \ge \tfrac{1}{k} \sum_{j \in [p]} |\beta_j|.$$

At the same time, we also need to satisfy $|\beta_1| \le \phi_1$ from the first majorization constraint. We now discuss two cases

$\diamond$ **Case 1:** If $\frac{1}{k} \sum_{j \in [p]} |\beta_j| \ge |\beta_1|$, in order to solve the minimization problem in (20), we set $\phi_1 = \frac{1}{k} \sum_{j=1}^n |\beta_j|$. Notice that $\phi_1 \le M$ is automatically satisfied because $\phi_1 = \frac{1}{k} \sum_{j \in [p]} |\beta_j| = \frac{1}{k} \sum_{j \in [p]} M z_j \le M$. This leads to $\phi_2 = \ldots = \phi_k = \frac{1}{k} \sum_{j \in [p]} |\beta_j|$. To see this, for the sake of contradition, assume that $\exists j \in \{2, \ldots, k\}$ such that $\phi_j < \frac{1}{k} \sum_{j \in [p]} |\beta_j|$. Since $\phi_j \le \phi_1 = \frac{1}{k} \sum_{j \in [p]} |\beta_j|$, we have $\sum_{j \in [k]} \phi_j < \sum_{j \in [k]} \frac{1}{k} \sum_{j \in [p]} |\beta_j| = \sum_{j \in [p]} |\beta_j|$, which contradicts the majorization constraint.

$\diamond$ **Case 2:** If $\frac{1}{k} \sum_{j \in [n]} |\beta_j| < |\beta_1|$, we can set $\phi_1 = |\beta_1|$. Notice that $\phi_1 \le M$ is automatically satisfied because $|\beta_1| \le M z_1 \le M$. Then we are left with $k - 1$ coefficients to set, and we can follow the same argument as we did for $j = 1$ with slight difference that the majorization constraints are changed to

$$\sum_{j=2}^l \phi_j \ge \sum_{j=2}^l |\beta_j| \quad \forall l \in \{2, \ldots, p-1\} \quad \text{and} \quad \sum_{j=2}^p \phi_j = \sum_{j=2}^p |\beta_j|.$$

We repeat this process until we set all $k$ coefficients $\phi_1, \ldots, \phi_k$, as implemented by Algorithm 2. The output of the algorithm coincides with the optimal value of the minimization problem in (20). Hence, the first claim follows.

As for the complexity claim, it is easy to see that Algorithm 2. only requires partial sorting step on Line 2, which has a complexity of $\mathcal{O}(p \log k)$. The summation step on Line 3 has a complexity of $\mathcal{O}(p)$. The for-loop step on Line 4-8 has a complexity of $\mathcal{O}(k)$, so does the final summation step on Line 9. Therefore, the overall computational complexity of Algorithm 2 is $\mathcal{O}(p + p \log k)$. This concludes the proof. $\square$

## Appendix B. Experimental Setup Details

### B.1. Setup for Evaluating Proximal Operators

The synthetic data generation process is as follows. We sample the input vector $\gamma \in \mathbb{R}^p$ from the standard multivariate Gaussian distribution, $\gamma \sim \mathcal{N}(\mathbf{0}, \boldsymbol{I}_p)$, where $\boldsymbol{I}_p$ denotes the identity matrix with dimension $p$. We vary the dimension $p \in \{2^0, 2^1, ..., 2^{10}\} \times 10^2$ and set the cardinality $k$ to be 10, the box constraint $M$ to be 1.0, and the weight parameter $\rho$ to be 1.0. We report the running time for evaluating these proximal operators. To obtain the mean and standard deviation of the running time, we repeat each setting 5 times, each with a different random seed.

### B.2. Setup for Solving the Perspective Relaxation

We generate our synthetic datasets in the following procedure. First, we sample each feature vector $\boldsymbol{x}_i \in \mathbb{R}^p$ from a Gaussian distribution, $\boldsymbol{x}_i \sim \mathcal{N}(\mathbf{0}, \boldsymbol{\Sigma})$, where the covariance matrix has entries $\Sigma_{jl} = \sigma^{|j-l|}$. The variable $\sigma \in (0,1)$ controls the features correlation: if we increase $\sigma$, feature columns in the design matrix $\boldsymbol{X}$ become more correlated. Throughout the experimental section, we set $\sigma = 0.5$. Next, we create the sparse coefficient vector $\boldsymbol{\beta}^*$ with $k$ equally spaced nonzero entries, where $\beta_j^* = 1$ if $j \bmod (p/k) = 0$ and $\beta_j^* = 0$ otherwise. After these two steps, we build the prediction vector $\boldsymbol{y}$. If our loss function is squared error loss (regression task), we set $y_i = \boldsymbol{x}_i^T \boldsymbol{\beta}^* + \epsilon_i$, where $\epsilon_i$ is a Gaussian random noise with $\epsilon_i \sim \mathcal{N}(0, \frac{\|\boldsymbol{X}\boldsymbol{\beta}^*\|}{\text{SNR}})$, and SNR stands for the signal-to-noise ratio. In all our experiments, we choose SNR $= 5$. If our loss function is logistic loss (classification task), we set $y_i \sim Bern(\boldsymbol{x}_i^T \boldsymbol{\beta}^* + \epsilon_i)$, where $Bern(P)$ is a Bernoulli random variable with $\mathbb{P}(y_i = 1) = P$ and $\mathbb{P}(y_i = -1) = 1 - P$. For this experiment, we vary the feature dimension $p \in \{1000, 2000, 4000, 8000, 16000\}$. We control the sample size by using a parameter called $n$-to-$p$ ratio, or sample to feature ratio. For the results in the main paper, we set $n$-to-$p$ ratio to be 1.0, the box constraint $M$ to be 2, the number of nonzero coefficients k (also the cardinality constraint) to be 10, and $\ell_2$ regularization coefficient $\lambda_2$ to be 1.0. Again, we report and compare the running times, with means and standard deviations calculated based on 5 repeated simulations with different random seeds.

### B.3. Setup for Certifying Optimality

**Datasets and Preprocessing**   We run on both synthetic and real-world datasets. For the synthetic datasets, we run on the largest synthetic instances ($n = 16000$ and $p = 16000$). For the real-world datasets, we use the dataset cancer drug response (Liu et al., 2020) for linear regression and DOROTHEA (Asuncion & Newman, 2007) for logistic regression.

The cancer drug response dataset has 822 samples and orginally has 34674 features. However, many feature only has a single value, so we prune all these features, which result in 2200 features. The DOROTHEA dataset originally has 1150 samples and 100000 features. However, the dataset is highly imbalanced (only 112 samples for the minority class). To alleviate this data imbalance problem and train a meaningful classifier, we apply the resampling preprocessing technique on the DOROTHEA dataset. We sample (with replacement) 1150 samples from each class. Thus, in total, we have 2300 samples. After pruning redundant features, we have 89989 features.

For both the cancer drug response and DOROTHEA dataset, we center each feature to have mean 0 and norm equal to 1.

**Choice of Hyperparameters**   For the cardinality constraint $k$, we set $k = 10$ for both synthetic datasets. This corresponds to the true number of nonzero coefficients. For the cancer drug response dataset, we set $k = 5$. For DOROTHEA, we set $k = 10$. The $k$ values for both real-world datasets are selected based on doing 5 fold cross validation with a heuristic sparse learning algorithm first. We apply the heuristics to get a path of solutions with different number of nonzero coefficients. The $k$ values are selected on this path of solutions right before the loss objective just starts to plateau or increase on the test set.

For the $\ell_2$ regularization coefficient, we set $\lambda_2 = 1$. For the box constraint, we set $M = 2$ for the synthetic datasets and DOROTHEA. The infinity norm of the final optimal solution less than this value. For the cancer drug response dataset, we set $M = 5$, which is also bigger than the infinity norm of the final optimal solution.

**Branch and Bound**   We first provide some background on the branch and bound (BnB) framework and then elaborate on the details of our BnB implementation. Problem 1 is both nonconvex and NP-hard. Thus, any method capable of certifying optimality, including heuristics, branching, bounding (calculating lower bounds), presolving, among many others. Our paper focuses on rapidly solving relaxation problems at both the root and node levels. This, in turn, provides fast lower bound certificates for the BnB algorithm.

For our method, we write a customized BnB framework. We use Algorithm 3 to solve the relaxation at each node and use Equation (13) to calculate the safe lower bound to prune the search space. To find feasible solutions, we use an effective approach called beamsearch (Liu et al., 2022) from the existing literature. For branching, we branch on the feature based on the best feasible solution found by the beamsearch algorithm at each node. For the nonzero coefficients of this solution, we branch on the variable which would lead to the largest loss increase if the coefficient to $0$. The intuition is that such a variable is important and should be branched early in the BnB framework.

### B.4. Computing Platforms

When investigating how much GPU can accelerate our computation, we ran the experiments with both CPU and GPU implementations on the Nvidia A100s. For everything else, we ran the experiments with the CPU implementation on AMD Milan with CPU speed 2.45 Ghz and 8 cores. We conducted the experiments in the the Delta system at the National Center for Supercomputing Applications from the Advanced Cyberinfrastructure Coordination Ecosystem: Services & Support (ACCESS) program (Boerner et al., 2023).

# Appendix C. Additional Numerical Results

## C.1. Perturbation Study regarding Solving the Perspective Relaxation

### C.1.1. PERTURBATION STUDY ON $M$ VALUES

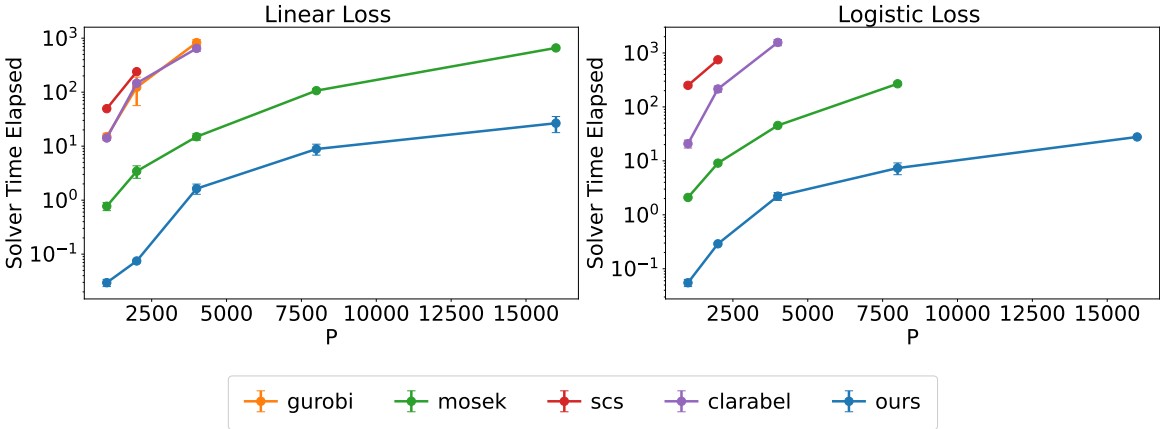

*Figure 4.* Solve the perspective relaxation in Problem (5). We set $M = 1.2$, $\lambda_2 = 1.0$, $n/p = 1$, and $k = 10$.

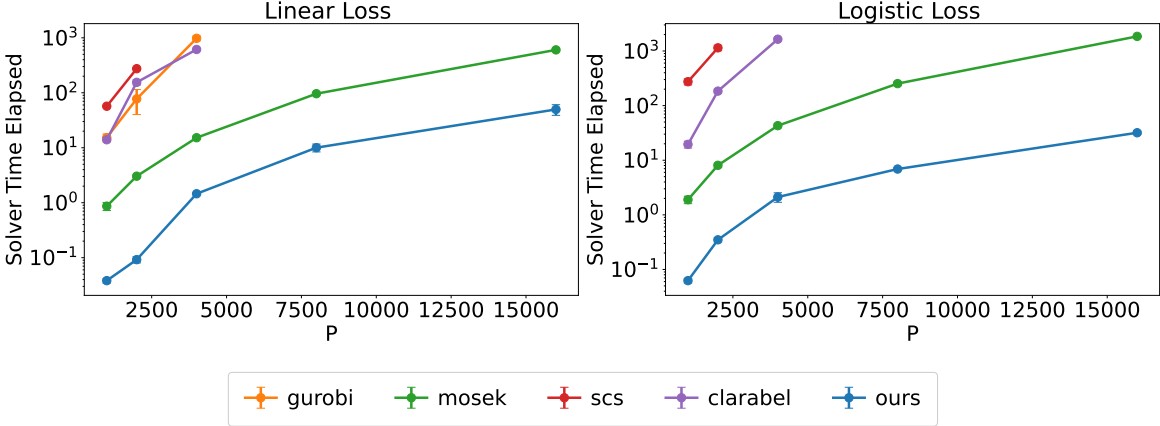

*Figure 5.* Solve the perspective relaxation in Problem (5). We set $M = 1.5$, $\lambda_2 = 1.0$, $n/p = 1$, and $k = 10$.

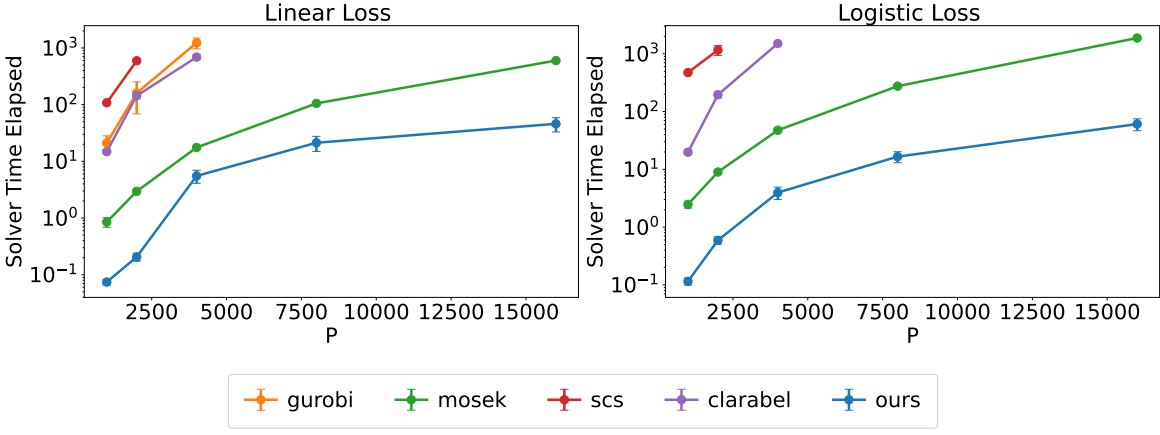

*Figure 6.* Solve the perspective relaxation in Problem (5). We set $M = 3.0$, $\lambda_2 = 1.0$, $n/p = 1$, and $k = 10$.

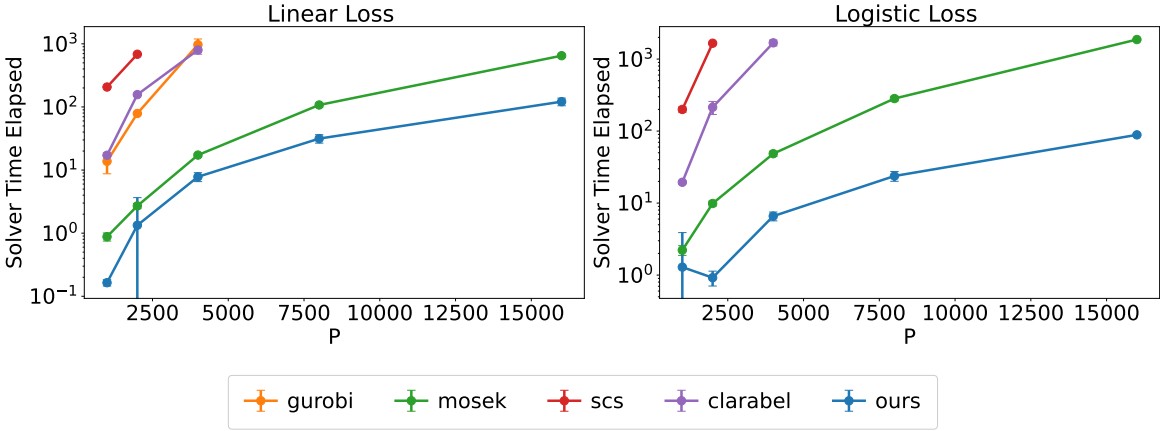

*Figure 7.* Solve the perspective relaxation in Problem (5). We set $M = 5.0$, $\lambda_2 = 1.0$, $n/p = 1$, and $k = 10$.

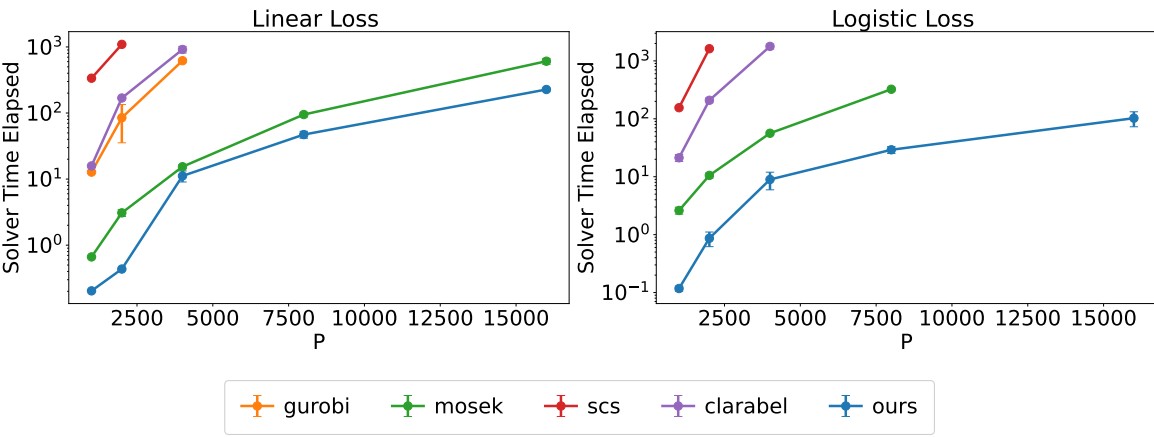

*Figure 8.* Solve the perspective relaxation in Problem (5). We set $M = 10.0$, $\lambda_2 = 1.0$, $n/p = 1$, and $k = 10$.

## C.1.2. Perturbation Study on $\lambda_2$ Values

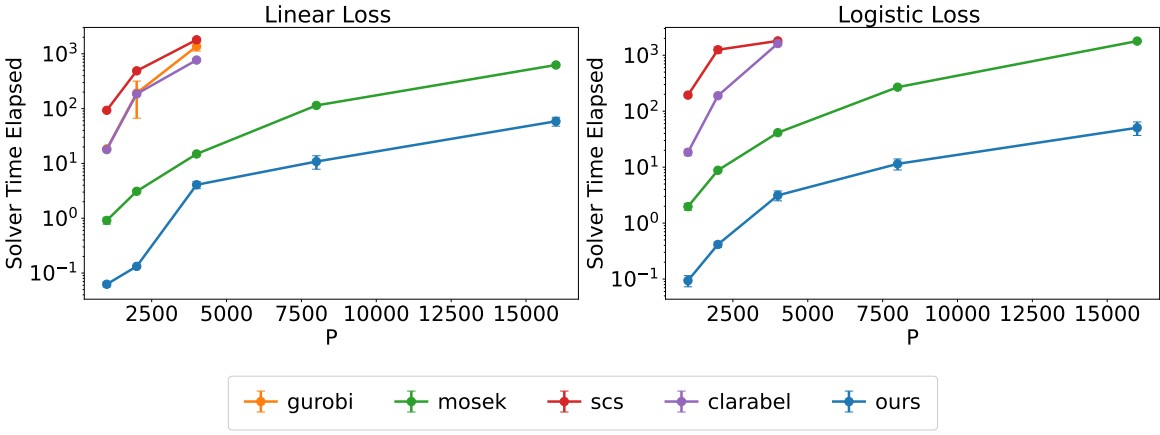

*Figure 9.* Solve the perspective relaxation in Problem (5). We set $M = 2.0$, $\lambda_2 = 0.1$, $n/p = 1$, and $k = 10$.

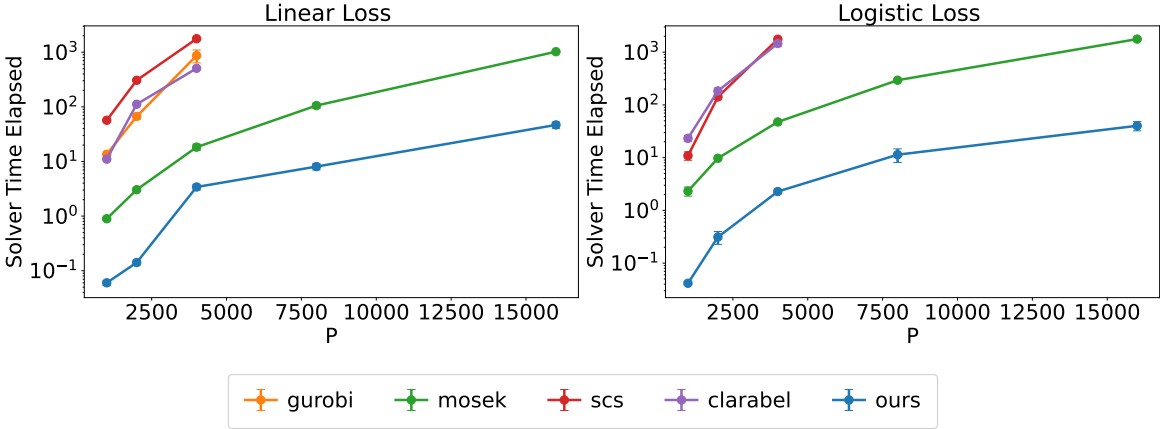

*Figure 10.* Solve the perspective relaxation in Problem (5). We set $M = 2.0$, $\lambda_2 = 10.0$, $n/p = 1$, and $k = 10$.

### C.1.3. PERTURBATION STUDY ON $n$-TO-$p$ RATIOS

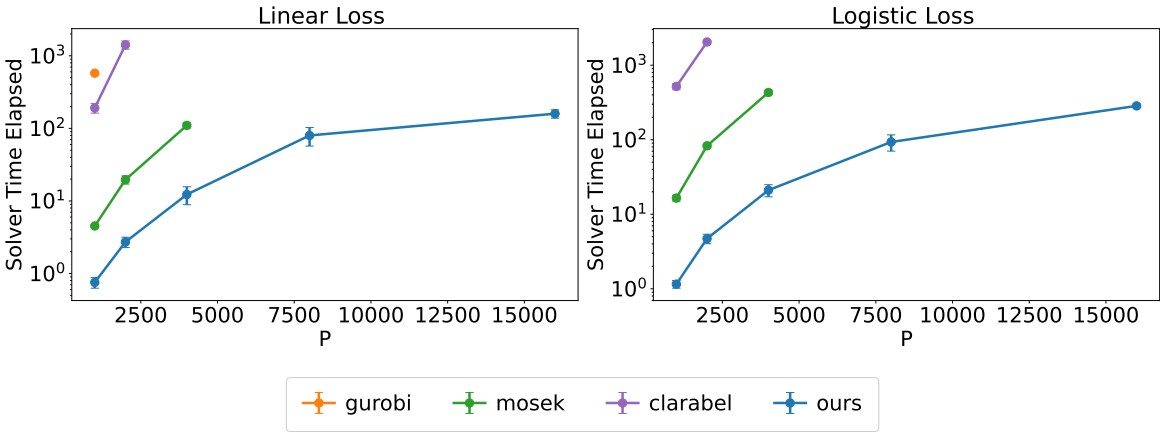

*Figure 11.* Solve the perspective relaxation in Problem (5). We set $M = 2.0$, $\lambda_2 = 1.0$, $n/p = 10.0$, and $k = 10$.

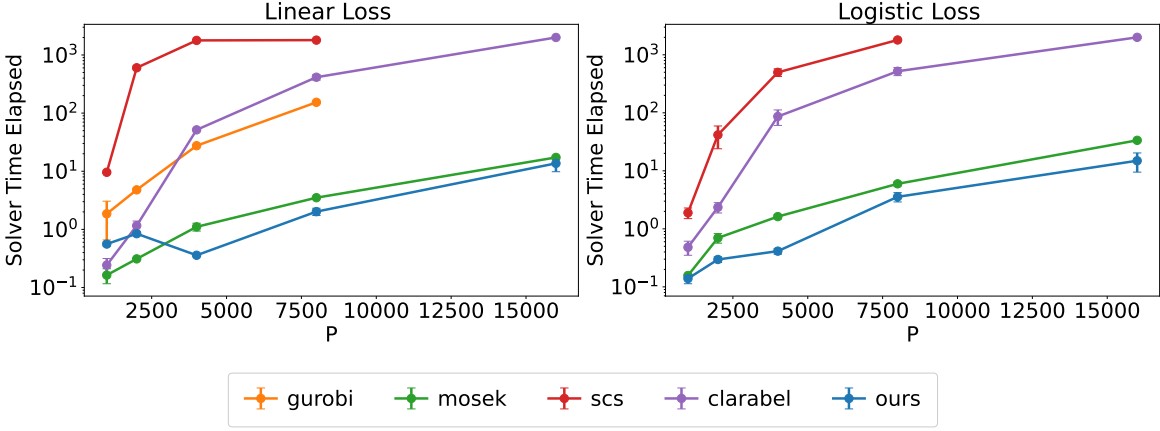

*Figure 12.* Solve the perspective relaxation in Problem (5). We set $M = 2.0$, $\lambda_2 = 1.0$, $n/p = 0.1$, and $k = 10$.

## C.1.4. PERTURBATION STUDY ON LARGE $k$ VALUE

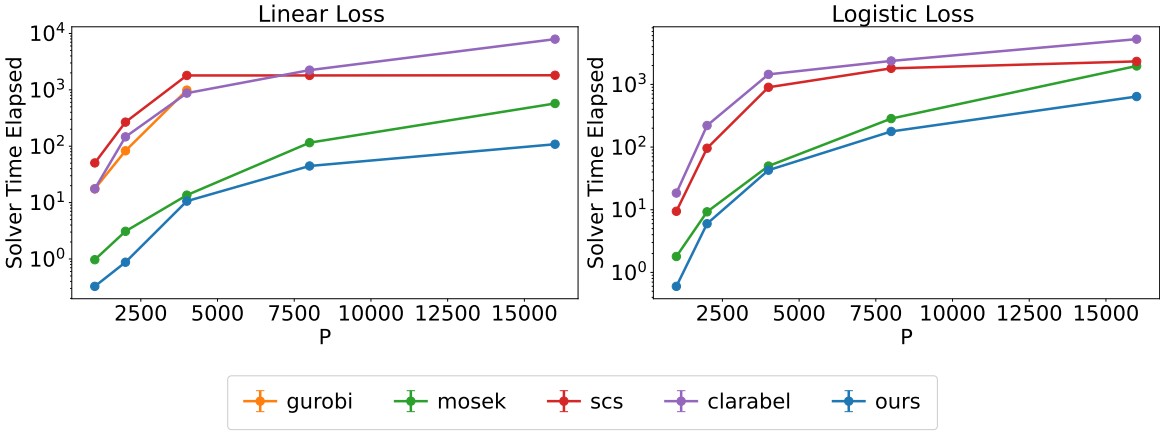

*Figure 13.* Solve the perspective relaxation in Problem (5). We set $M = 2.0$, $\lambda_2 = 1.0$, $n/p = 0.1$, and $k = 500$.

## C.2. Comparison between Perspective Relaxation and $\ell_1$-Relaxation

Besides the perspective relaxation discussed in Section 2, there is another common relaxation for the sparse learning problem, which is the $\ell_1$-relaxation.

Recall that the perspective relaxation problem is formulated as:

$$P^\star_{\text{conv}} = \begin{cases} \min\limits_{\boldsymbol{\beta}, \boldsymbol{z} \in \mathbb{R}^p} & f(\boldsymbol{X}\boldsymbol{\beta}, \boldsymbol{y}) + \lambda_2 \sum_{j \in [p]} \beta_j^2 / z_j \\ \text{s.t.} & \boldsymbol{z} \in [0,1]^p, \, \mathbf{1}^T \boldsymbol{z} \leq k, \\ & -Mz_j \leq \beta_j \leq Mz_j \, \forall j \in [p]. \end{cases} \tag{21}$$

In contrast, the $\ell_1$-relaxation problem is formulated as (following Mhenni et al. (2020, Equation 2)):

$$P^\star_{\text{conv}} = \begin{cases} \min\limits_{\boldsymbol{\beta}, \boldsymbol{z} \in \mathbb{R}^p} & f(\boldsymbol{X}\boldsymbol{\beta}, \boldsymbol{y}) + \lambda_2 \|\boldsymbol{\beta}\|_2^2 \\ \text{s.t.} & \|\boldsymbol{\beta}\|_1 \leq kM, \, \|\boldsymbol{\beta}\|_\infty \leq M. \end{cases} \tag{22}$$

Since Lemma 2.1 establishes that the perspective relaxation is equivalent to the convex hull of the $\ell_2$-regularization term subject to the cardinality and box constraints, this is the tightest convex relaxation for the $\ell_2$ term. Thus, the perspective relaxation is a better choice than the $\ell_1$-relaxation. We verify the superiority of the perspective relaxation over the $\ell_1$-relaxation by numerically comparing the two relaxation methods in terms of the objective values.

*Table 3.* Linear Regression Results: objective values of the perspective relaxation and $\ell_1$-relaxation (the higher the better).

| method | p=1000 | p=2000 | p=4000 | p=8000 | p=16000 |
|---|---|---|---|---|---|
| l1 | 1088.78 | 2418.70 | 5518.70 | 11877.03 | 25710.79 |
| perspective | 1119.53 | 2449.34 | 5549.40 | 11907.39 | 25741.67 |

*Table 4.* Logistic Regression Results: objective values of the perspective relaxation and $\ell_1$-relaxation (the higher the better).

| method | p=1000 | p=2000 | p=4000 | p=8000 | p=16000 |
|---|---|---|---|---|---|
| l1 | 202.12 | 485.82 | 1000.84 | 2216.15 | 4667.79 |
| perspective | 234.34 | 518.99 | 1032.95 | 2248.64 | 4699.78 |

Lastly, we want to mention that there exist theoretically tigher relaxations in the literature (particularly rank-one SOCP and SDP formulations), but these formulations face fundamental computational limitations. They require interior-point methods, which cannot be warm-started effectively and also scale poorly with problem size. Our perspective relaxation provides the optimal practical balance: it delivers sufficiently tight bounds while remaining computationally efficient through first-order methods.

# Appendix D. Additional Discussions

We first provide common calculus rules for conjugate functions, whose proof can be found in standard optimization textbooks such as (Beck, 2017).

◇ **Separable Sum Rule:** Let $f(\boldsymbol{x}) = \sum_{j \in [p]} f_j(x_j)$, where $f_j : \mathbb{R} \to \mathbb{R}$ is convex for all $j \in [p]$. Then, the conjugate of $f$ is given by $f^*(\boldsymbol{\mu}) = \sum_{j \in [p]} f_j^*(\mu_j)$.

◇ **Scalar Multiplication Rule:** Let $g : \mathbb{R}^p \to \mathbb{R}$ be convex and $\alpha > 0$ be a scalar. Then, the conjugate of $f(\boldsymbol{x}) = \alpha g(\boldsymbol{x})$ is given by $f^*(\boldsymbol{\mu}) = \alpha g^*(\boldsymbol{\mu}/\alpha)$.

◇ **Addition to Affine Function Rule:** Let $g : \mathbb{R}^p \to \mathbb{R}$ be convex and $\boldsymbol{a}, \boldsymbol{b} \in \mathbb{R}^p$ be two vectors. Then, the conjugate of $f(\boldsymbol{x}) = g(\boldsymbol{x}) + \boldsymbol{a}^\top \boldsymbol{x} + b$ is given by $f^*(\boldsymbol{\mu}) = g^*(\boldsymbol{\mu} - \boldsymbol{a}) - \boldsymbol{b}$.

◇ **Composition with Invertible Linear Mapping Rule:** Let $g : \mathbb{R}^p \to \mathbb{R}$ be convex and $\boldsymbol{A} \in \mathbb{R}^{p \times p}$ be an invertible matrix. Then, the convex conjugate of $f(\boldsymbol{x}) = g(\boldsymbol{Ax})$ is given by $f^*(\boldsymbol{\mu}) = g^*(\boldsymbol{A}^{-\top}\boldsymbol{\mu})$.

◇ **Infimal Convolution Rule:** Let $g, h : \mathbb{R}^p \to \mathbb{R}$ be convex. Then, the convex conjugate of $f(\boldsymbol{x}) = \inf_{by} g(\boldsymbol{y}) + h(\boldsymbol{x} - \boldsymbol{y})$ is given by $f^*(\boldsymbol{\mu}) = g^*(\boldsymbol{\mu}) + h^*(\boldsymbol{\mu})$.

These rules are useful for discussions in D.1 and D.2.

## D.1. Convex Conjugate for GLM Loss Functions

The convex conjugates of some of GLM loss functions are summarized bellow.

◇ **Linear Regression:**
$$F(\boldsymbol{X\beta}) = \|\boldsymbol{X\beta} - \boldsymbol{y}\|_2^2 \quad \& \quad F^*(-\boldsymbol{\zeta}) = \frac{1}{4}\|\boldsymbol{\zeta}\|_2^2 - \boldsymbol{y}^T\boldsymbol{\zeta}.$$

◇ **Logistic Regression:**
$$F(\boldsymbol{X\beta}) = \sum_{i \in [n]} \log(1 + \exp(-y_i(\boldsymbol{X\beta})_i)) \quad \& \quad F^*(-\boldsymbol{\zeta}) = \sum_{i \in [n]} \left(1 - \frac{\zeta_i}{y_i}\right) \log\left(1 - \frac{\zeta_i}{y_i}\right) + \frac{\zeta_i}{y_i} \log\left(\frac{\zeta_i}{y_i}\right).$$

◇ **Poisson Regression:**
$$F(\boldsymbol{X\beta}) = \sum_{i \in [n]} \left(\exp(\boldsymbol{X\beta})_i - y_i(\boldsymbol{X\beta})_i\right) \quad \& \quad F^*(-\boldsymbol{\zeta}) = \sum_{i \in [n]} h(-\zeta_i + y_i),$$

where $h(z) = z \log(z) - z$ if $z > 0$ and $h(z) = 0$ if $z = 0$.

◇ **Gamma Regression:**
$$F(\boldsymbol{X\beta}) = \sum_{i \in [n]} \left(y_i \exp(-(\boldsymbol{X\beta})_i) + (\boldsymbol{X\beta})_i\right) \quad \& \quad F^*(-\boldsymbol{\zeta}) = \sum_{i \in [n]} y_i h(\frac{1 - \zeta_i}{y_i}),$$

where $h(z) = z \log(z) - z$ if $z > 0$ and $h(z) = 0$ if $z = 0$.

◇ **Squared Hinge Loss:** For binary classification with labels $y_i \in \{-1, +1\}$,
$$F(\boldsymbol{X\beta}) = \sum_{i \in [n]} \max(0, 1 - y_i(\boldsymbol{X\beta})_i)^2 \quad \& \quad F^*(-\boldsymbol{\zeta}) = \sum_{i \in [n]} h(-y_i\zeta_i),$$

where $h(z) = z + \frac{z^2}{4}$ if $z \leq 0$ and $h(z) = \infty$ if $z > 0$.

◇ **Multinomial Logistic Regression:** For multiclass classification with $K$ classes with coefficients $\boldsymbol{\beta} \in \mathbb{R}^{p \times K}$, let $y_{ik}$ be a binary indicator such that $y_{ik} = 1$ if the $i$-th sample belongs to class $k$, and $y_{ik} = 0$ otherwise.

$$F(\boldsymbol{X\beta}) = \sum_{i \in [n]} \left( \log \left( \sum_{j=1}^{K} \exp((\boldsymbol{X\beta})_{ij}) \right) - \sum_{k=1}^{K} y_{ik} (\boldsymbol{X\beta})_{ik} \right) \quad \& \quad F^*(-\boldsymbol{\zeta}) = \sum_{i \in [n]} h(\boldsymbol{y}_i - \boldsymbol{\zeta}_i),$$

where $h(\boldsymbol{z}) = \sum_{k=1}^{K} z_k \log(z_k)$ if $\boldsymbol{z} > \boldsymbol{0}$ and $\boldsymbol{1}^T \boldsymbol{z} = 1$, and $h(\boldsymbol{z}) = \infty$ otherwise.

### D.2. Safe Lower Bound

The linear regression problem with eigen-perspective relaxation is formulated as

$$P^\star_{\text{eig-conv}} = \min_{\boldsymbol{\beta} \in \mathbb{R}^p} \boldsymbol{\beta}^\top \boldsymbol{Q}_{\text{eig}} \boldsymbol{\beta} - 2\boldsymbol{y}^\top \boldsymbol{X} \boldsymbol{\beta} + 2\lambda_{\text{eig}} g(\boldsymbol{\beta}),$$

where $\boldsymbol{Q}_{\text{eig}} = \boldsymbol{X}^\top \boldsymbol{X} - \lambda_{\min}(\boldsymbol{X}^\top \boldsymbol{X}) \boldsymbol{I}$, $\lambda_{\text{eig}} = \lambda_2 + \lambda_{\min}(\boldsymbol{X}^\top \boldsymbol{X})$, and $\lambda_{\min}(\cdot)$ denotes minimum eigenvalue of the input matrix. Using the standard version of weak duality theorem, we have

$$P^\star_{\text{MIP}} \geq P^\star_{\text{eig-conv}} \geq -F^*(-\hat{\boldsymbol{\zeta}}) - G^*(\hat{\boldsymbol{\zeta}}),$$

where $F(\boldsymbol{\beta}) = \boldsymbol{\beta}^\top \boldsymbol{Q}_{\text{eig}} \boldsymbol{\beta}$, $G(\boldsymbol{\beta}) = -2\boldsymbol{y}^\top \boldsymbol{X} \boldsymbol{\beta} + 2\lambda_{\text{eig}} g(\boldsymbol{\beta})$, and $\hat{\boldsymbol{\zeta}} = -\nabla F(\hat{\boldsymbol{\beta}}) = -2\boldsymbol{Q}_{\text{eig}} \hat{\boldsymbol{\beta}}$. The conjugate functions admit the following closed form expressions

$$F^*(-\hat{\boldsymbol{\zeta}}) = \frac{1}{4} \hat{\boldsymbol{\zeta}}^\top \boldsymbol{Q}_{\text{eig}}^\dagger \hat{\boldsymbol{\zeta}} = \hat{\boldsymbol{\beta}} \boldsymbol{Q}_{\text{eig}} \hat{\boldsymbol{\beta}} \quad \& \quad G^*(\hat{\boldsymbol{\zeta}}) = 2\lambda_{\text{eig}} \, g^* \left( \frac{-\boldsymbol{Q}_{\text{eig}} \hat{\boldsymbol{\beta}} + \boldsymbol{X}^\top \boldsymbol{y}}{\lambda_{\text{eig}}} \right),$$

where we use $(\cdot)^\dagger$ to denote the pseudo-inverse of a matrix. We may conclude that

$$P^\star_{\text{MIP}} \geq \hat{\boldsymbol{\beta}} \boldsymbol{Q}_{\text{eig}} \hat{\boldsymbol{\beta}} + 2\lambda_{\text{eig}} \, g^* \left( \frac{-\boldsymbol{Q}_{\text{eig}} \hat{\boldsymbol{\beta}} + \boldsymbol{X}^\top \boldsymbol{y}}{\lambda_{\text{eig}}} \right).$$

The above lower bound can be viewed as a generalization of the safe lower bound formula from (Liu et al., 2024, Theorem 3.1). Specifically, as $M$ approaches $\infty$, the above lower bound matches the lower bound in in (Liu et al., 2024, Theorem 3.1). Furthermore, Our proof uses a simple weak duality argument and is concise, in contrast to the lengthy two-page algebraic proof of (Liu et al., 2024, Theorem 3.1).

