# OpenReview forum: "Scalable First-order Method for Certifying Optimal k-Sparse GLMs"
_ICML.cc/2025/Conference — ICML 2025 poster_

### Official Review · Reviewer_xg9D · 2025-03-08

**Overall Recommendation:** 3

**Summary:**

In this work, the authors proposed a novel FISTA-based algorithm for computing a lower bound in the Branch-and-Bound method. The new algorithm utilizes several customized components and outperforms universal optimization solvers on both artificial and practical datasets. Please see the following sections for my detailed comments.

**Claims And Evidence:**

Due to the time limit, I did not check the correctness of the theory, except those briefly mentioned in the main paper. The theoretical claims seem correct by checking the main paper.

**Essential References Not Discussed:**

N/A.

**Experimental Designs Or Analyses:**

The experiment settings and results are sound and reasonable to me.

**Methods And Evaluation Criteria:**

The methods and evaluation criteria make sense to me.

**Other Comments Or Suggestions:**

- Algorithm 3, Line 5: should the step size be $\phi / (\phi + 3)$?

**Other Strengths And Weaknesses:**

Please see my comments in other sections.

**Questions For Authors:**

Please see my comments in other sections.

**Relation To Broader Scientific Literature:**

This paper is related to the topic of ICML conference and should be interesting to audiences from machine learning and optimization fields.

**Theoretical Claims:**

Due to the time limit, I did not check the correctness of the theory. The theoretical claims and the proofs in the main paper seem correct.

---

> ### Author Rebuttal · Authors · 2025-04-01
>
> We thank the reviewer for carefully reading our paper! This is indeed a typo. The implementation is correct in our submitted source code. We will fix this typo in the revision.

---

### Official Review · Reviewer_X5XP · 2025-03-13

**Overall Recommendation:** 2

**Summary:**

This paper explores the use of branch-and-bound (BnB) frameworks to solve sparsity-constrained optimization problems.

The authors derive a lower bound for the optimization problem using a perspective relaxation formulation.

To efficiently solve the resulting perspective relaxation, the authors employ a first-order proximal gradient algorithm.

Extensive experiments on both synthetic and real-world datasets demonstrate that the proposed approach accelerates dual bound computations and improves efficiency.

**Claims And Evidence:**

yes

**Essential References Not Discussed:**

yes

**Experimental Designs Or Analyses:**

yes

**Methods And Evaluation Criteria:**

The parameter $k$, set to $k \in $\{5,10,15\}$, is too small.

**Other Comments Or Suggestions:**

No

**Other Strengths And Weaknesses:**

### **Strengths:**
- **S1.** The proposed method significantly accelerates computations compared to previous BnB algorithms.
- **S2.** The authors derive the proximal operator of the function $g(\beta)$ in problem (7). The authors demonstrate that the customized PAVA algorithm in Algorithm 1 has an $\tilde{O}(p)$ time complexity.  These results are reasonable to me.


### **Weaknesses:**
- **W1.** The main novelty of the paper is unclear. The perspective formulation is not new, the BnB framework and the APG algorithm are well-established, and the Pool Adjacent Violators Algorithm (PAVA) has already been used for solving isotonic regression problems and SLOPE models.

- **W2.** The justification for using the perspective formulation to estimate the lower bound is not well explained.

- **W3.** The Customized PAVA algorithm in Algorithm 1 appears to be an incremental extension of the standard PAVA algorithm. Its design is primarily based on leveraging the property that the optimal solution of problem (11) maintains the same order as the input data
$\mu$. While the algorithm introduces some new ideas, its overall contribution seems limited.

- **W4.** The parameter $k$ used in the experiments is too small, set to $k \in \{5,10,15\}$. With such small values, standard local algorithms such as OMP, iterative hard thresholding, BCD methods would likely perform well. This raises concerns about whether the proposed method offers a meaningful improvement in real-world applications.

**Questions For Authors:**

- **Q1.** How does the scalability of the proposed method change when $k = 0.1p$?

**Relation To Broader Scientific Literature:**

Most of the results are empirical, with a lack of theoretical analysis.

**Theoretical Claims:**

yes

---

> ### Author Rebuttal · Authors · 2025-04-01
>
> We thank the reviewer for thoughtful feedbacks.
>
> 1. **k is not large enough**
>
> a. *Many real-world datasets are naturally sparse*. Small $k$'s are sufficient for accurate prediction and can help avoid overfitting, especially on the validation set. Small $k$ also improves interpretability. This is true for the two real-world datasets used in the paper. We can follow the experimental setup section (Line 973-977) to select the correct $k$ via cross-validation. The best $k$ for Cancer Drug Response is $5$ (already used), and the best $k$ for DOROTHEA is $26$ (in the submission, we used $15$). Below is the BnB runtime comparison when $k=26$ for DOROTHEA. Our method is still significantly faster than MOSEK.
>
> |    | MOSEK  |  ours  |
> |:---:|:---:|:---:|
> | running time (s) | 661.33 | 228.34 |
>
> b. *Some data science practitioners require $k$ to be small*. This is the case for the sparse identification of dynamical systems  (Bertsimas & Gurnee, 2023). Physicists often want the differential equation to be succinct in order to match some physical intuition or prior knowledge.
>
>
> c. Lastly, *our lower bound calculation is still fast when $k$ is large*. We re-ran Fig. 2 (main paper) with $k=500$ (runtime in seconds):. We only compare with MOSEK as other three baselines are already struggling when $p=4000$.
>
> Linear regression:
> | method  |  p=1000   |  p=2000   |  p=4000   |  p=8000   |  p=16000   |
> | :-----: |  :-----:  |  :-----:  |  :-----:  |  :-----:  |  :-----:   |
> | MOSEK   |  0.970    |  3.097    |  13.565   |  115.554  |  573.247   |
> | ours    |  0.327    |  0.876    |  10.623   |  44.638   |  108.475   |
>
> Logistic regression:
>
> | method  |   p=1000   |  p=2000   |  p=4000   |  p=8000   |  p=16000   |
> | :-----: |   :-----:  |  :-----:  |  :-----:  |  :-----:  |   :-----:  |
> | MOSEK   |   1.794    |  9.266    |  49.801   |  284.638  |  1969.305  |
> | ours    |   0.601    |  5.993    |  42.802   |  177.481  |   640.345  |
>
> 2. **Novelty is unclear: perspective formulation, BnB algorithm, APG algorithm, and PAVA are not new**
>
> We clarify the distinction between prior work and our contributions:
>
> a. *BnB*: Our key innovation lies not in using BnB but in developing an efficient 1st order method for computing the BnB lower bounds.
>
> b. *Perspective relaxation*: Again, the novelty is not using the perspective relaxation but in solving it.
>
> c. *APG*: Although the APG framework is standard, its direct application to our problem is impossible without our method to evaluate the proximal operator efficiently and exactly.
>
> d. *PAVA*: Novelty is not using PAVA but linking $\text{prox}_{\rho^{-1}g}(\cdot)$ with PAVA via generalized isotonic regression reformulation and then invoking the Moreau decomposition Theorem.
>
> Additional contributions include:
>
> a. Lemma 3.1 shows $g^*$ is a simple composition of the Huber loss and the top-$k$ norm.
>
> b. Our Algorithm 2 computes $g(\beta ^{t+1})$ exactly and efficiently, crucial for restart.
>
> c. Our lower bound computation is GPU-friendly
>
> 3. **Justification for perspective formulation is not clear**
>
> Our main goal is to certify optimality, which necessitates using a BnB framework, whose efficiency depends strongly on obtaining tight lower bounds.
> Our perspective relaxation produces tighter lower bounds than traditional relaxation approaches (such as the $\ell_1$ relaxation in our Q1 to Reviewer enX8).
>
> 4. **Greedy/heuristic algorithms**
>
> We completely agree that heuristics are very effective.
> The heuristic method used in our work is already a variant of OMP (Line 983-989; also see our source code) and can sometimes find the optimal solution even within a few seconds (the whole BnB algorithm can still take over hundreds of seconds though).
> However, this is not the focus of this work and orthogonal to our goal, which is to certify optimality.
> Our contributions lies in efficiently computing the lower bounds at each node.
>
> 5. **Meaningful real-world applications for certifying optimality**
>
> Please see Section 1.2. Related Works for more applications.
> We highlight a few below:
>
> a. *Sparse identification of dynamical systems (Bertsimas \& Gurnee, 2023)*: The datasets contain highly-correlated features, on which solving to optimality can greatly outperform heuristic methods.
>
> b. *Medical scoring systems (Ustun \& Rudin, 2019)*: Certifying optimality ensures that physicians provide the best disease diagonostic system to the patients.
>
> c. *Portfolio optimization (Bienstock, 1996)*: Solving the problem to optimality ensures that companies are picking the optimal financial trading strategy.
>
> We sincerely appreciate your thoughtful questions and engagement with our work.
> We hope to have answered to your concerns and questions and we would be happy to provide further explanations.

---

### Official Review · Reviewer_UQGX · 2025-03-13

**Overall Recommendation:** 2

**Summary:**

This paper studies sparse generalized linear models with cardinality constraints. Existing branch-and-bound methods are not computationally efficient due to expensive or slow dual bound computations. To overcome this, the authors propose a first-order proximal gradient algorithm to solve the perspective relaxation of the problem. The method eliminates the need for costly second-order cone programming and incorporates a restart strategy to accelerate convergence. Experiments demonstrate improvements in computation.

**Claims And Evidence:**

The title of the paper is somewhat misleading, as the proposed method primarily relies on a convex relaxation rather than directly addressing the original sparse ridge regression problem (1). Additionally, the claims about the optimality gap are supported only by numerical experiments, without theoretical justification.

**Essential References Not Discussed:**

Using GPUs to speed up sparse learning is not new. There is existing literature on this topic (see, e.g., [1]). Therefore, I believe some statements in this paper to be overly exaggerated.

- [1] Blanchard, Jeffrey D., and Jared Tanner. "GPU accelerated greedy algorithms for compressed sensing." Mathematical Programming Computation 5.3 (2013): 267-304.

**Experimental Designs Or Analyses:**

The experimental design does not consider scenarios where $k$ is large (e.g., $p/2$), especially when $p$ is also large.

**Methods And Evaluation Criteria:**

- The theoretical result in the paper only focus on the properties in computation while lacks of results in optimization. For example, what is the quality of solution?
- What is the empirical performance the method when applying the method to Poisson regression?

**Other Comments Or Suggestions:**

Typos:

- Table 1: p should be presented in a math form.
- "Alas" --> "Also"

**Other Strengths And Weaknesses:**

- Strengths
The empirical results show that the proposal is promising for reducing computational time.

- Weaknesses
  - I believe the special case of Lemma 3.1, where $M=\infty$, likely appears in existing literature, but this is not discussed in the paper.
  - The paper is not well-structured:
    - It is confusing about the problem the authors aim to solve.
    - Some information about FISTA should be presented before the Methodology section.
    - The assumptions in the paper are not clearly stated.

**Questions For Authors:**

Please refer to **Claims and Evidence**, **Methods and Evaluation Criteria**, and **Weaknesses**.

**Relation To Broader Scientific Literature:**

The key contributions of this paper align with the broader literature on optimization with sparsity constraints. Thus, it may be helpful for high-dimensional data analysis.

**Theoretical Claims:**

I have reviewed Appendix A and examined the proof of Lemma 3.1 in detail. While the proof appears to be correct, some assumptions are not clearly stated or justified in the paper.

---

> ### Author Rebuttal · Authors · 2025-04-01
>
> We thank the reviewer for the thoughtful feedback.
>
> We briefly restate our contribution to avoid any misunderstanding.
> Our paper addresses certifying optimality or quantifying the optimality gap for the $\ell_0$-constrained GLMs, \textbf{without making any assumptions on the data}.
> Our motivation stems from the need for optimal solutions in high-stakes applications, where solving the $\ell_0$ problem ensures superior quality, interpretability, and trustworthiness compared to heuristics.
> However, the problem is NP-hard, necessitating a BnB algorithm.
> This BnB framework consists of several key components, including heuristics, branching, bounding (calculating lower bounds), pre-solving, among many others.
> The effectiveness of BnB methods strongly depends on the quality and speed of solving its continuous relaxation (lower bound).
> In this work, we have proposed a first-order method to solve it efficiently. By doing so, we have significantly expanded the size/scale of the datasets on which the BnB approach can be successfully applied.
>
> With this explanation and context, we can now answer the questions raised by the reviewer.
>
>
> 1. **Confusion about the problem to solve**
>
> Our contribution is tailored at efficiently computing the lower bound for the $\ell_0$-constrained GLMs as the main workhorse for solving the full problem by BnB.
> We hope that by adjustments to the paper (and the title, see below) this confusion is removed.
>
> 2. **Optimality is supported without theoretical justification**
>
> It should be now clear that optimality is guaranteed by the use of the BnB algorithm.
>
> 3. **Title is misleading**
>
> Hopefully we have clarified the reviewer's misunderstanding and explained why the title represents our goal and contribution well.
> However, we are open to discuss an alternative title along the line of ``Certifying Optimality of k-Sparse GLMs by Scalable First-order Lower Bound Computation".
>
> 4. **Some assumptions are not clearly stated or justified**
>
> We do not impose any assumption on our datasets. However, if the reviewer thinks there are specific parts of paper that require more explicit discussion, we are open to follow them up to improve the clarity of the paper.
>
> 5. **Cite Blanchard and Tanner**
>
> Thank you for this reference. We will cite this work in revision. While both works use GPU acceleration, they address fundamentally different problems.
> The cited work focuses on accelerating greedy/heuristic methods through GPUs.
> There is no requirement of proving optimality.
> In contrast, our work harnesses GPU specifically for computing lower bounds in BnB, which is crucial for pruning nodes and certifying optimality.
> By relying solely on matrix-vector multiplications rather than solving linear systems, we achieve the first truly GPU-efficient implementation for computing the BnB's lower bounds.
>
>
> 6. **$k$ is not large enough**
>
> Due to space limit, please see our reply (Q1) to Reviewer X5XP.
>
> 7. **Poisson regression**
>
> We run the setup in Figure 2 (main paper) with Poisson regression for MOSEK, ours, and ours+GPU.
> The running time (in seconds) are reported below. "***" denotes reaching the time limit (1800s).
>
> | method   |  p=1000   |  p=2000   |  p=4000   |  p=8000   |  p=16000   |
> | :-----:  |  :-----:  |  :-----:  |  :-----:  |  :-----:  |  :-----:   |
> | MOSEK |  2.537    |  15.505    |  136.790   |  919.412   |   ***  |
> | ours CPU  |  2.505   |  12.821    |  95.583   |  572.876  |  ***   |
> | ours GPU |  2.637 | 6.142 |  5.933   |  17.338   |  25.954   |
>
> The GPU version of our method scales very well on large-scale instances.
>
> 8. **Cite more papers when $M=\infty$**
>
> Due to space limit, please see our reply (Q1) to Reviewer 3khc. In revision, we will cite those two papers and discuss the key differences.
>
> 9. **More background information on FISTA**
>
> Thank you for the suggestion. In revision, we will include a section in the appendix covering the FISTA algorithm and the Nesterov acceleration method.
>
> 10. **usage of  alas**
>
> Thank you very much. We will change ''alas'' to make clear that we meant the sentence in adversative form, i.e., we will use ''however'' in the revision.
>
> We sincerely appreciate your thoughtful questions and engagement with our work.
> We hope to have answered to your concerns and questions and we would be happy to provide further explanations.

---

### Official Review · Reviewer_3khC · 2025-03-14

**Overall Recommendation:** 4

**Summary:**

The paper proposes a new algorithm for solving lower bounds in the BnB framework. It solves a composite problem using an efficient restarted FISTA algorithm, for which an efficient way to exactly compute the function value and the proximal operator is given. As such, the paper allows to achieve impressive efficiency gains compared to existing methods.

**Claims And Evidence:**

The authors prove their main claims or give appropriate references when necessary (Section 2 and 3), and provide extensive experiments to show the advantage of their method (Section 4)

**Essential References Not Discussed:**

This is just a suggestion, but I think it might be interesting to cite the papers [1] or [2] below related to the k-support norm. Indeed, the function $g^{\*}$ in the paper is very similar to the (half-squared) top k-norm (the top-k norm is the norm of the top-k elements in absolute value), which is the dual function of the (half-squared) k-support norm (since the top-k norm is the dual norm of the k-support norm). More precisely, for a given $\boldsymbol{\alpha}$, if all for all $i$, $|\alpha_i| \leq M$, then $g^*(\boldsymbol{\alpha})$ is exactly the half-squared top-k norm.
Now, the proximal operator of the k-support norm is known [1] and can be efficiently computed from [2]: therefore I wonder if a similar technique could be used to give an alternative computation for the proximal operator of $g^*$ (related to the one of $g$). Indeed, Algorithm 1 seems to involve different techniques, with the merging of intervals. In any case, I notice that in [2], the complexity of the proximal operator is in $O(p log(p))$, which is the same as the proximal operator in this reviewed paper (assuming $\tilde{O}(p)$ in the paper means $O(p \log(p))$, not $O(p   \text{poly}(\log(p))$ for some higher order polynomial ? Maybe the authors can confirm). Therefore, I think this reviewed paper gives a very efficient formulation, and therefore this remark is more just for sake of curiosity rather than a suggestion for improving the algorithm.

[1] Sparse Prediction with the k-Support Norm, Argyriou et al.

[2] New Perspectives on k-Support and Cluster Norms, McDonald et al.

**Experimental Designs Or Analyses:**

The experiments are synthetic experiments and their setting makes sense to me.

**Methods And Evaluation Criteria:**

The setting of the paper makes a lot of sense: it builds on existing methods (BnB methods using lower bounds based on relaxations, to solve sparse optimization problems), and replaces the part of the lower bound computation by an efficient FISTA algorithm using a novel proximal operator.

**Other Comments Or Suggestions:**

See questions below.

**Other Strengths And Weaknesses:**

I think this paper uses ingenious techniques to make the computation of lower bounds in BnB much more effective. The paper uses innovative techniques for the computation of the proximal operator, and I think the restart technique is also welcomed, to improve the efficiency of FISTA: that last technique actually makes a big difference as the authors show in their experiments.
The overall complexity of the algorithm is shown to be much better than state of the art solvers like Gurobi, which is very encouraging.

**Questions For Authors:**

I first have just a small question regarding the comparison with the k-support norm, although the authors can omit it as it is just for sake of curiosity rather than a suggestion of improvement.

And I would also have another second question regarding the performance of the full algorithm plugged in the BnB framework, to solve (1). Indeed, I think it would be interesting to compare the complexity or running time of solving the full optimization problem (1) with the method in the paper (BnB with lower bounds based on the improved FISTA) vs. with other methods. Can we already say that the method in the paper would also be more efficient than other methods for solving (1) ? More precisely: this paper provides a more efficient first order method to compute lower bounds for BnB. In the introduction, it is said such first order methods are usually slow, but give tighter bounds than methods based e.g. on conic relaxations. Now, since the method in the paper is faster, can we say that overall (i.e. for solving the full (1)), it is a better methods than all other alternatives ? Under which conditions is it ? (Note: I am less familiar with the literature on BnB so feel free to react if my question needs to be reformulated/updated)

**Relation To Broader Scientific Literature:**

This paper give a more efficient way to solve sparse optimization problems, which is very useful in many areas of machine learning such as medical scoring systems or portfolio optimization, as described in the paper's introduction.

**Theoretical Claims:**

I did not check the correctness of the proofs in details, but I am familiar with similar literature on the computation of proximal operators for sparse penalties, and the results make sense to me at first sight.

---

> ### Author Rebuttal · Authors · 2025-04-01
>
> We thank the reviewer for their thoughtful feedback. We address their concerns below:
>
> 1. **Connection to Argyriou et al. and McDonald et al.**
>
> Thank you for these valuable references--we will cite and discuss both papers.
> We fully agree with your observations.
> We clarify some key differences:
>
> **Difference 1** Our function $g(\beta)$ generalizes the k-support norm. Specifically, when  $M=\infty$, our formulation reduces exactly to the standard k-support norm. However, our work makes several novel contributions beyond prior works:
>
>    a. We are the first to study the Fenchel dual of $g$
>
>    b. We derive an explicit closed-form expression for $g^*$ (Equation 9)
>
>    c. Our analysis captures bounded variable constraints ($ M < \infty $)
>
> **Difference 2** From a computational perspective, ours differs fundamentally from prior works:
>
>    a. Algorithmic Framework: Both cited papers compute the proximal operator directly in the primal space (their Algorithm 1's), while our paper employs a dual approach by computing the proximal operator of the dual function $g^*$ and applying the Moreau decomposition theorem (Equation 12)
>
>    b. Constraint Handling: To the best of our knowledge, the primal approach struggles to incorporate box constraints effectively, whereas our dual framework naturally accommodates box constraints while maintaining computational efficiency.
>
>    c. Implementation Advantages: The dual perspective provides a more flexible computational framework, enabling efficient handling even for constrained optimization scenarios
>
> 2. **Does $\tilde{O}(p)$ mean $O(p \log(p))$?**
>
> Yes, this is correct.
> In Algorithm 1, the most expensive step is the sorting step (line 3), which has computational cost $O(p \log(p))$.
> All other steps have computational costs $O(p)$.
> Thus, the overall computational cost is $O(p \log(p))$.
>
> 3. **Is our method (BnB with lower bounds computed by our Algorithm 3) better than all other methods in solving the overall Problem 1? Under what conditions?**
>
> To properly address this question, let us first provide some context about BnB algorithms. Problem (1) is both nonconvex and NP-hard.
> Thus, any method capable of certifying optimality, including ours, must ultimately rely on BnB. This BnB framework consists of several key components, including heuristics, branching, bounding (calculating lower bounds), presolving, among many others.
> Our paper focuses on rapidly solving relaxation problems at both the root and node levels. This, in turn, provides fast lower bound certificates for the BnB algorithm.
>
> With this background information, we can now more directly address your 1st question. The answer is yes in the sense that it is currently the method that achieves the best results and scalability thanks to efficient lower bound computation. The key advantage lies in our formulation's compatibility with first-order optimization methods, which enables warm-starting and maintains computational tractability even for large-scale problems.
> While there exist theoretically tighter relaxations in the literature (particularly rank-one SOCP and SDP formulations), these formulations face fundamental computational limitations. They require interior-point methods, which cannot be warm-started effectively and also scale poorly with problem size. Our perspective relaxation provides the optimal practical balance: it delivers sufficiently tight bounds while remaining computationally efficient through first-order methods.
>
> For your 2nd question, our method performs the best when applied to **large-scale** problems, on which existing lower bound computation methods scale poorly. In such cases, our approach provides substantial improvements in both speed and scalability. For small size problems, we observe that computing lower bounds is generally computationally cheap. In these scenarios, both our branch-and-bound implementation and commercial mixed-integer programming solvers can typically certify optimality within seconds. While these smaller cases serve as useful validation of our method's correctness, they are not the primary focus of our work, as they do not present the same computational challenges that motivate our technical contributions.
>
> We sincerely appreciate your thoughtful questions and engagement with our work.
> We hope to have answered to your concerns and questions and we would be happy to provide further explanations.

---

### Official Review · Reviewer_enX8 · 2025-03-14

**Overall Recommendation:** 4

**Summary:**

This paper considers the problem of sparse generalized linear model (GLM) optimization. Specifically, the goal is to fit a generalized linear model under the constraint that at most $k$ features are used. Typical approaches for such problem include LASSO, greedy/local search-based techniques, as well as branch-and-bound approaches.

The authors focus on developing a fast algorithm for solving the convex relaxation that appears in each step of the branch-and-bound process. While the usual relaxation employed (e.g. in LASSO) replaces the $\ell_0$ sparsity constraint by an $\ell_1$ constraint, this relaxation is not necessarily tight. Instead, the authors employ convex duality to study the convex hull of the $\ell_0$-based discrete optimization set, and end up with a tighter but non-closed form constraint. This constraint essentially regularizes the top (largest-magnitude) features separately from the bottom ones, using an $\ell_2$ norm for the former but an $\ell_1$ norm for the latter. The authors show how to evaluate as well as optimize over this regularizer efficiently, leading to an overall efficient algorithm for solving the relaxation. Experiments show that this specialized algorithm outperforms off-the-shelf second-order cone programming solvers.

Overall, I believe this is a good paper that provides a thorough and valid analysis of the sparse GLM problem. I believe it could be used to further move the state of the art by making branch-and-bound-based sparsity solvers more efficient. This would be significant for quality-sensitive applications.

**Claims And Evidence:**

I found all the claims presented to be valid.

**Essential References Not Discussed:**

N/A

**Experimental Designs Or Analyses:**

N/A

**Methods And Evaluation Criteria:**

The method is evaluated on synthetic tasks, using both CPU and GPU, as well as two real linear/logistic regression datasets, which are highly over-parametrized. The experiments are performed multiple times to reduce variance. The results show a significant benefit of using the proposed method (the version with ranodm restarts) compared to off-the-shelf optimizers like gurobi.

**Other Comments Or Suggestions:**

- Add some comparison with ISTA.
- An optional suggestion is to take a look at papers in the optimal transport / deep learning literature for similar approaches regarding a soft-top-k operator. https://arxiv.org/pdf/2002.06504 https://arxiv.org/pdf/2304.04947 At a high level they have the same goal of inducing tighter relaxations to the $\ell_0$ operator.

**Other Strengths And Weaknesses:**

The paper is well-written.

The box constraint parameter $M$ is introduced but then ignored in Algorithm 2 (I guess it assumes $M=\infty$). It might also be implicitly assumed in other algorithms / theorems. It would be good for the authors to fix this.

**Questions For Authors:**

- How do the results compare to ISTA (i.e. no acceleration)?
- Is there any experimental evidence that the relaxation used here is superior to the $\ell_1$-based relaxation? It would be great to see at least one experiment or some pointers to other works that do that.

**Relation To Broader Scientific Literature:**

The idea of solving sparse optimization with branch-and-bound has been studied before, e.g. "Sparse Branch and Bound for Exact Optimization of L0-Norm Penalized Least Squares" by Mhenni et al. However that work uses an $\ell_1$ relaxation. I am not aware of any work using a tighter relaxation.

**Theoretical Claims:**

All the theoretical claims look correct to me. I checked the proof of Theorem 3.6 and all the claims in the main body.

---

> ### Author Rebuttal · Authors · 2025-04-01
>
> We thank the reviewer for their thoughtful feedback. We address their concerns below:
>
> 1. **Connection to Mhenni et al. and experimental comparison**
>
> Thank you; we will cite this paper. However, we note that our perspective relaxation is tighter than the $\ell_1$ relaxation from by Mhenni et al.
> Specifically, Lemma 2.1 establishes that the closed convex hull of
>
> \begin{align}
> \\{ ( \beta,  z, t): ||\beta||_2^2 \leq t, z \in \\{0, 1\\}^p, 1^T z \leq k,  ||\beta_j||  _{\infty}\leq M  \\}
> \end{align}
> is
>
> \begin{align}
> \\{ ( \beta, z, t) : \sum _{j=1}^p \beta_j^2/z_j \leq t, z \in [0,1]^p, 1^\top z \leq k, -M z_j \leq \beta_j \leq M z_j ~~ \forall j \in [p] \\}.
> \end{align}
> This is the tightest possible relaxation for the $\ell_2$ term.
>
> The perspective relaxation we want to solve within the BnB algorithm is:
> \begin{align}
> \min_{\beta, z} || y - X \beta || _2^2 + \lambda_2 \sum _{j=1}^p \beta_j^2/z_j \quad \text{s.t.} \quad z \in [0, 1]^p, 1^T z \leq k, -M z_j \leq \beta_j \leq M z_j
> \end{align}
>
> In contrast, the $\ell_1$-based relaxation proposed by Mhenni et al. (Equation 2 in their paper and subsequent discussion) will solve:
> \begin{align}
> \min_{\beta, z} || y - X \beta ||_2^2 + \lambda_2 ||\beta||_2^2 \quad \text{s.t.} \quad ||\beta||_1 \leq k M, ||\beta|| _{\infty} \leq M
> \end{align}
>
> As Lemma 2.1 indicates, our perspective relaxation provides tighter lower bounds, crucial for pruning nodes and certifying optimallity.
> Below we provide some numerical comparison (same setup as Figure 2 in the paper):
>
> Linear regression (the higher the better):
> | method      | p=1000  | p=2000  | p=4000  | p=8000   | p=16000  |
> | :-----:     | :-----: | :-----: | :-----: | :-----:  | :-----:  |
> | l1          | 1088.78 | 2418.70 | 5518.70 | 11877.03 | 25710.79 |
> | perspective | 1119.53 | 2449.34 | 5549.40 | 11907.39 | 25741.67 |
>
> Logistic regression (the higher the better):
> | method      |  p=1000  | p=2000  | p=4000  | p=8000  | p=16000 |
> | :-----:     |  :-----: | :-----: | :-----: | :-----: | :-----: |
> | l1          |  202.12  | 485.82  | 1000.84 | 2216.15 | 4667.79 |
> | perspective |  234.34  | 518.99  | 1032.95 | 2248.64 | 4699.78 |
>
> 2. **Does Algorithm 2 only work when $M=\infty$?**
>
> Algorithm 2 is valid for any $M > 0$.
> In Algorithm 3, we first compute the proximal update (line 8), before evaluating $g(\beta ^{t+1})$ in Algorithm 2.
> Since $\beta ^{t+1}$ is the output of the proximal operator, it inherently satisfies all constraints in Equation (7).
> Namely, there exists some $z \in [0, 1]^p$ such that $1^T z \leq k$ and $-M z_j \leq \beta_j \leq M z_j \; \forall j \in [p]$.
> This property is critical to the proof of Theorem 3.6 (Line 896-897 and Line 902-903), where we show that Algorithm 2 correctly computes $g(\beta ^{t+1})$.
> We will clarify and emphasize this point in revision.
>
> 3. **Comparison with ISTA**
>
> If by ISTA the reviewer refers to the non-accelerated version of FISTA, the comparison is already included in Figure 3, as PGD (Proximal Gradient Descent) and ISTA are essentially the same algorithm. We used the more general term PGD since ISTA typically implies an $\ell_1$ regularizer. We would be happy to rename legend to ''PGD/ISTA'' in revision. Please let us know if we have misinterpreted your meaning regarding ISTA.
>
> 4. **Connection to optimal transport/deep learning literature for soft-top-k operator**
>
> Thank you; we will cite these two papers.
>
> However, the smooth approximation of the top-k operator is not suitable here.
> The effectiveness of the proximal algorithm relies on the exact evaluation of the proximal operator. Replacing the top-k would lead to solving a different problem rather than the proximal operator evaluation.
> This will not help us to use FISTA to solve the perspective relaxation. As a result, this approach would not guarantee valid lower bounds necessary for optimality certification.
>
>
> We hope to have answered to your concerns and questions, and we would be happy to provide further explanations.

---

### Decision · Program_Chairs · 2025-05-01

**Decision:**

Accept (poster)

**Comment:**

The paper presents a first-order proximal gradient algorithm for optimizing lower bounds within a BnB framework for solving the $k$-sparse GLM problem. The paper received five detailed reviews. Three reviews are in favor of accepting this work, while the other two lean towards rejection. While the opinions expressed by the reviewers are somewhat mixed, the majority opinion is that the proposed FISTA-type lower bound optimizer in BnB is innovative, and it works to considerably reduce the overall computation cost of non-convex sparse GLM.  The reviewers also raised some concerns regarding the inadequate references and missing theoretical/empirical comparisons with some related works, which have been thoroughly addressed by authors in their responses. Based on the reviews and author responses, it was assessed that this submission makes a sufficiently novel and solid contribution to the addressed problem, and thus may be accepted. The authors are encouraged to address the issues raised in the reviews as much as possible in the revised paper.